# Structured illumination with particle averaging reveals novel roles for yeast centrosome components during duplication

**Shannon Burns[1], Jennifer S Avena[2], Jay R Unruh[1], Zulin Yu[1], Sarah E Smith[1], Brian D Slaughter[1], Mark Winey[2], Sue L Jaspersen[1,3]***

[1]Stowers Institute for Medical Research, Kansas City, United States; [2]Department of Molecular, Cellular and Developmental Biology, University of Colorado Boulder, Boulder, United States; [3]Department of Molecular and Integrative Physiology, University of Kansas Medical Center, Kansas City, United States

**Abstract** Duplication of the yeast centrosome (called the spindle pole body, SPB) is thought to occur through a series of discrete steps that culminate in insertion of the new SPB into the nuclear envelope (NE). To better understand this process, we developed a novel two-color structured illumination microscopy with single-particle averaging (SPA-SIM) approach to study the localization of all 18 SPB components during duplication using endogenously expressed fluorescent protein derivatives. The increased resolution and quantitative intensity information obtained using this method allowed us to demonstrate that SPB duplication begins by formation of an asymmetric Sfi1 filament at mitotic exit followed by Mps1-dependent assembly of a Spc29- and Spc42-dependent complex at its tip. Our observation that proteins involved in membrane insertion, such as Mps2, Bbp1, and Ndc1, also accumulate at the new SPB early in duplication suggests that SPB assembly and NE insertion are coupled events during SPB formation in wild-type cells.

***For correspondence:** slj@stowers.org

**Competing interests:** The authors declare that no competing interests exist.

## Introduction

The goal of molecular, biochemical and cell biological studies is to elucidate the function of cellular components and understand how proteins and the complexes they form interact in vivo. Few methods are available to examine large protein structures such as microtubule-organizing centers (MTOCs) in intact cells. However, elucidating the mechanism of MTOC assembly is important since MTOCs perform a variety of functions in the cell, including cell signaling, cilia assembly, intraflagellar transport, and chromosome segregation. Duplication of a class of MTOCs known as centrosomes (in metazoans) or spindle pole bodies (SPBs, in fungi) once per cell cycle is essential to facilitate the formation of a bipolar spindle (*Winey and O'Toole, 2001*; *Winey and Bloom, 2012*). Defects in centrosome duplication are linked to cancer and other diseases (*Chavali et al., 2014*; *Godinho and Pellman, 2014*).

Perhaps the best characterized MTOC both cytologically and molecularly is the SPB of *Saccharomyces cerevisiae*. Electron microscopy (EM) studies show that the SPB is a multilayered cylindrical organelle that is embedded in the nuclear envelope (NE) throughout the yeast lifecycle (*Byers and Goetsch, 1974*, *1975*). The outer and inner plaques of the SPB nucleate cytoplasmic and nuclear microtubules, respectively, while the central plaque is key to the structural integrity of the SPB and anchorage of the SPB to the NE via hook-like appendages visible by electron tomography (*O'Toole et al., 1999*). Associated with one side of the SPB is a modified region of the NE known as

**eLife digest** Cells divide to produce two new daughter cells that each contain the same genetic material. First, the DNA of the parent cell is copied, then it must be physically separated into the daughter cells by a structure made of filaments called microtubules. To ensure that the DNA is separated into two equal parts, the microtubules must emerge from two points in the cell, known as spindle poles.

Each spindle pole is made of a group (or 'complex') of proteins and these have to be copied before the cell can divide. While we understand how DNA is copied, we do not know how cells copy proteins. The spindle pole in yeast—known as the spindle pole body—is an ideal model to study this problem because the proteins that form it have already been identified and it is easy to study yeast in the laboratory.

Burns et al. developed a new method to study the spindle pole body using fluorescent protein tags and a sophisticated microscopy technique. The experiments mapped the positions of 18 proteins within the spindle pole body during its duplication. Some of these proteins enable the spindle pole to insert into the membrane that surrounds the cell's nucleus. Unexpectedly, Burns et al. observed that this set of proteins interact with the new spindle pole as it forms, instead of afterwards as was previously believed.

Burns et al.'s findings suggest that the spindle pole body assembles into the membrane surrounding the nucleus at the same time as it is copied. The next challenges are to understand the details of how this works and to use the same method to study other large protein complexes in cells. Until now, highly detailed surveys of protein structures have been limited to a handful of proteins and conditions. The method developed by Burns et al. makes it possible to carry out studies that examine the movements of whole protein complexes during cell division.

the half-bridge, which is important for SPB duplication and cytoplasmic microtubule formation during G1 phase and mating (*Byers and Goetsch, 1974*, *1975*; *Knop and Schiebel, 1998*). Although SPBs and centrosomes are morphologically distinct, they share a number of components, including proteins involved in duplication (Sfi1 and Cdc31/centrin), structure (Spc110/pericentrin and Nud1/centriolin), microtubule nucleation (γ-tubulin complex, composed of Tub4, Spc97, and Spc98 in budding yeast), regulators (Mps1 and cyclin-dependent kinases; Cdks), and assembly principles (*Jaspersen and Winey, 2004*; *Kollman et al., 2011*; *Winey and Bloom, 2012*; *Fu et al., 2015*). Analysis of SPB duplication in yeast has provided key mechanistic and regulatory insights into the control of centrosome duplication in higher eukaryotes (*Fu et al., 2015*).

Formation of a new SPB occurs adjacent to the mother SPB. Duplication is typically broken down into three steps based on cytological examination of wild-type and mutant yeast cells (*Figure 1A*) (*Adams and Kilmartin, 2000*; *Jaspersen and Winey, 2004*; *Winey and Bloom, 2012*). A combination of molecular, genetic, and cytological methods has been used to determine the composition and distribution of the 18 SPB components within the organelle and to predict their function(s) in its duplication (*Figure 1B*). The integral membrane proteins Mps3 and Kar1 localize to the nuclear and cytoplasmic face of the half-bridge, respectively, and likely serve as structural components that are involved in the localization of Sfi1 and Cdc31 (*Vallen et al., 1992a*, *1994*; *Spang et al., 1995*; *Jaspersen et al., 2002*). Sfi1 is a long helical protein that contains multiple binding sites along its length for the yeast centrin, Cdc31 (*Kilmartin, 2003*; *Li et al., 2006*). Dephosphorylation of Cdk1 sites in the C-terminus of Sfi1 initiates elongation of the half-bridge to a full bridge, most likely via oligomerization of C-terminal ends of Sfi1 in an anti-parallel manner that allows free N-termini to associate directly or indirectly with components of the satellite—the new SPB precursor (*Avena et al., 2014*; *Elserafy et al., 2014*). ImmunoEM predicts that the satellite is composed of four of the 18 SPB components, Spc42, Spc29, Cnm67, and Nud1 (*Adams and Kilmartin, 1999*). How the satellite assembles is unknown, but within the mother SPB core, trimers of Spc42 dimers form a hexagonal lattice that is visible by cryoEM (*Bullitt et al., 1997*). The N-terminus of Spc42 associates with Spc29 and with Spc110, a spacer protein that tethers the γ-tubulin complex to the inner plaque (*Elliott et al., 1999*; *Muller et al., 2005*). The C-terminus of Spc42 associates with Cnm67, another spacer protein that associates with the signaling scaffold Nud1/centriolin and the outer plaque receptor for the

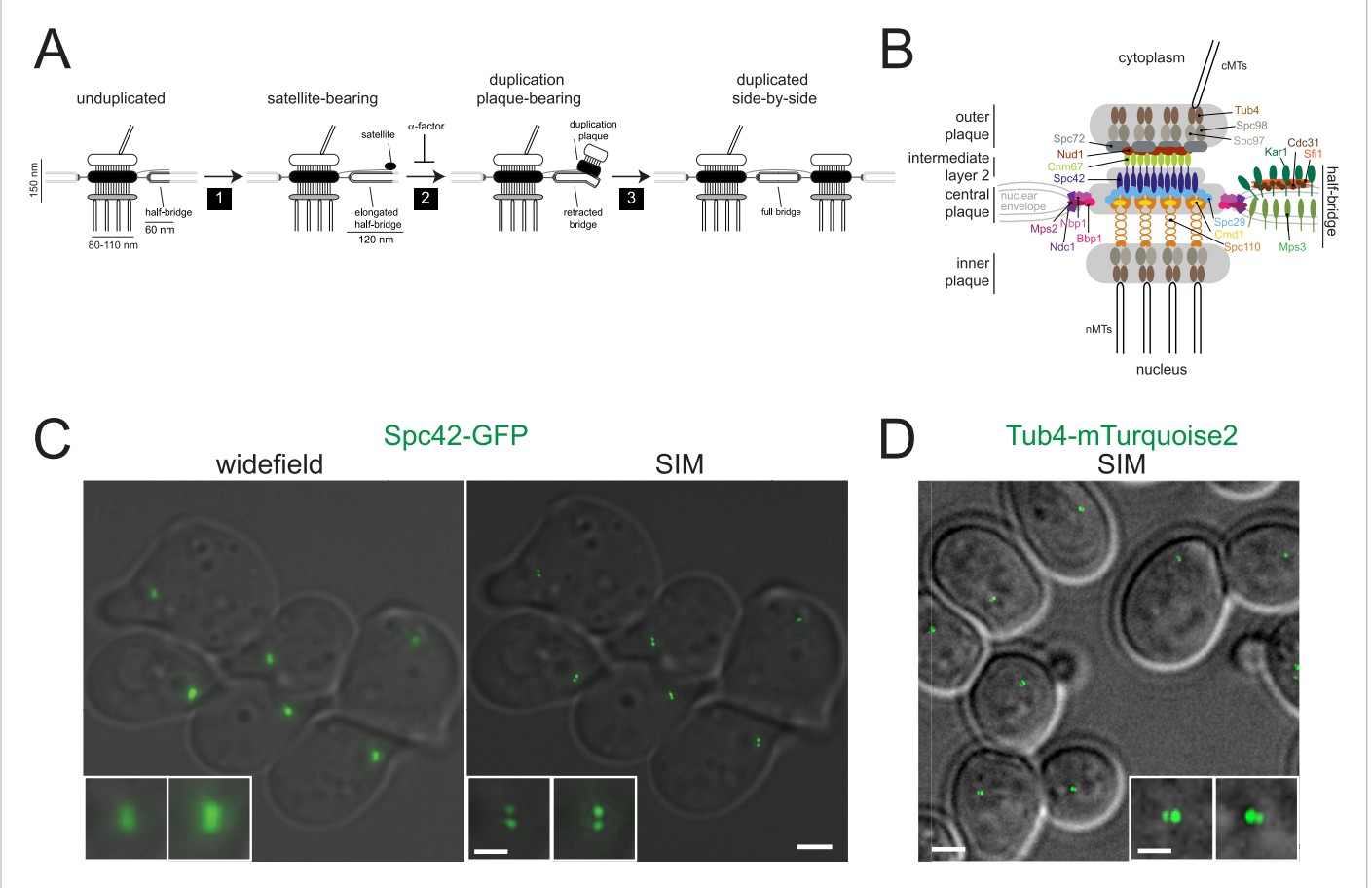

**Figure 1**. Spindle pole body (SPB) sub-structures can be visualized by structured illumination microscopy (SIM). (**A**) Schematic of the SPB duplication pathway deduced from electron microscopy (EM) analysis of wild-type and mutant yeast, including the size of SPB substructures (reviewed in *Adams and Kilmartin, 2000*; *Jaspersen and Winey, 2004*; *Winey and Bloom, 2012*). Three steps of SPB duplication: (1) elongation of the half-bridge and deposition of the satellite; (2) maturation of the satellite into a structure known as a duplication plaque and retraction of the bridge; (3) insertion of the duplication plaque into the NE and assembly of nuclear SPB components to create duplicated side-by-side SPBs. Treatment of cells with α-factor blocks SPB duplication at the satellite-bearing stage. (**B**) Schematic of the SPB showing the locations of all 18 components based on immunoEM, FRET, yeast two-hybrid, biochemical, and genetic data. (**C**) Comparison of widefield and SIM using Spc42-GFP, which is present in the core SPB and satellite in α-factor arrested cells. Insets show the SPB from top left and bottom right cells. (**D**) SIM of asynchronous Tub4-mTurquoise2 cells. Insets show SPBs from large-budded anaphase cell. Bars, 2 μm and 200 nm (inset).

γ-tubulin complex, Spc72 (*Knop and Schiebel, 1998*; *Gruneberg et al., 2000*). Linkage of the SPB to the NE is thought to occur via Spc29, which binds to the soluble protein Bbp1 that in turn associates with the integral membrane protein Mps2 (*Schramm et al., 2000*). However, other tethering mechanisms may exist. For example, mutations in *NDC1* and *NBP1* result in many of the same phenotypes as *BBP1* and *MPS2* mutants, including defects in SPB insertion into the NE (*Winey et al., 1991*; *Chial et al., 1998*; *Schramm et al., 2000*; *Araki et al., 2006*; *Kupke et al., 2011*; *Chen et al., 2014*). Therefore, it is presumed that Ndc1 and Nbp1 also anchor the SPB in the NE through unknown binding interactions with the non-membrane core SPB.

To understand the mechanism of SPB duplication and insertion into the NE, we applied structured illumination microscopy (SIM) and single particle averaging (SPA) to all 18 subunits of the SPB expressed at endogenous levels as fusions to green fluorescent protein (GFP), yellow fluorescent protein (YFP), mCherry (a red fluorescent protein derivative), cyan fluorescent protein (CFP), or mTurquoise2 (a CFP derivative). In contrast to SPA approaches used in previous studies with super-resolution microscopy data (*Loschberger et al., 2012*; *Szymborska et al., 2013*; *Loschberger et al., 2014*; *Van Engelenburg et al., 2014*; *Broeken et al., 2015*), we employed three-dimensional image

fitting of the protein intensity distributions to facilitate image alignment and intensity quantitation in our dual color images. SPA-SIM provided unexpected insights into the early steps of SPB duplication that were not attainable using existing technologies, including the structure and timing of half-bridge elongation, the composition of the satellite and the formation of the membrane pore. Relative spatiotemporal distributions of protein density from multiple images were obtained using fudicial markers and cell cycle analysis. We find that assembly of both the bridge and satellite occurs through a series of discrete and ordered steps, beginning with end-to-end association of Sfi1 in late mitosis. Interestingly, the Sfi1 filament is not symmetrical in the elongated half-bridge and there is relatively more Cdc31 bound to the newly assembled Sfi1. In addition to known satellite components, we find that Mps2 and Bbp1 localize to a region near the distal cytoplasmic tip of the extended half-bridge during SPB duplication, while Ndc1 is found along its length and Nbp1 is restricted primarily to the mother SPB. These data suggest that half-bridge and membrane proteins couple SPB duplication with its NE insertion.

## Results

### SPB structure and duplication can be studied using SIM

The small size of the budding yeast SPB (150 nm height, 80–110 nm diameter in haploids, 90–150 nm half-bridge length; [*Byers and Goetsch, 1974*; *Winey et al., 1991*; *Li et al., 2006*]; *Figure 1A,B*) falls below the ~200 nm resolution limit of conventional widefield and confocal microscopes. SIM provides a twofold increase in this resolution limit (*Gustafsson et al., 2008*), and we were able to detect two foci of Spc42-GFP that were unresolvable using widefield microscopy (*Figure 1C*). In most α-factor arrested cells, two Spc42 foci were observed; one focus was significantly brighter than the other, a phenotype anticipated for a mother SPB and a satellite that is beginning to assemble (*Figure 1A,C*). The distance between foci (225 ± 10 nm) was calculated from the center of the mother SPB to the center of the satellite so it is greater than the reported length of the extended half-bridge (117 ± 9 nm), which was measured by EM from the edge of SPB to the edge of the satellite (*Li et al., 2006*). Analysis of Spc42-mMaple by photo-activated localization microscopy (PALM) reported a distribution of distances between the mother SPB and satellite as 80–280 nm (*Seybold et al., 2015*). By SIM, two closely spaced foci of Tub4-mTurquoise2 were also detected at the spindle poles of most cells (*Figure 1D*). The average distance between foci was 157 ± 7 nm, which is similar to the 150 nm height measured from outer to inner plaque by EM (*Byers and Goetsch, 1974*; *Winey et al., 1991*). These results show that SPB substructures (the inner and outer plaques) and duplication intermediates (satellite) can be detected by SIM of endogenously expressed fluorescent fusion proteins, which was not possible with previous techniques.

### Sfi1 undergoes an asymmetric end-to-end fusion during half-bridge elongation

ImmunoEM analysis of Sfi1 N- and C-termini showed a distribution of gold particles along the length of the bridge with most N-terminal ends located adjacent to the mother SPB or satellite and most C-terminal ends located in the central region of the bridge. Based in part on this and on the observation that Sfi1 is able to form long (60–90 nm) filaments in vitro, John Kilmartin proposed that elongation of the half-bridge may occur through end-to-end association of Sfi1 C-termini; this would generate a free Sfi1 N-terminus that could seed satellite assembly (*Figure 2A*) (*Li et al., 2006*). Using C-terminally tagged Sfi1 (Sfi1-GFP) in combination with Spc42-mCherry, we observed a single Sfi1 focus in 97% of cells treated with α-factor (*Figure 2B,C*). In contrast, two Sfi1 foci were observed in 62% of N-terminally tagged Sfi1 (GFP-Sfi1) cells arrested under the same conditions (*Figure 2B,C*). The orientation of the mother SPB and satellite and the relatively low fluorescence intensity of GFP-Sfi1 compared with other SPB components were the primary reasons why we do not see two foci of GFP-Sfi1 in all cells. The distance between GFP-Sfi1 foci was 187 ± 1 nm, which is longer than estimates of bridge length obtained from EM studies (117 ± 9 nm; *Li et al., 2006*).

Three important and novel observations were made regarding the distribution of Sfi1 on the extended half-bridge. Unlike Spc42-mCherry intensity, which is greatest at the mother SPB and shows reduced levels at the satellite, many cells exhibit increased GFP-Sfi1 fluorescence at the distal end near the satellite compared with the proximal end (*Figure 2B,C*), suggesting asymmetry along the Sfi1 filament. The C-terminus of Sfi1 was noticeably displaced towards the cytoplasmic side of the bridge

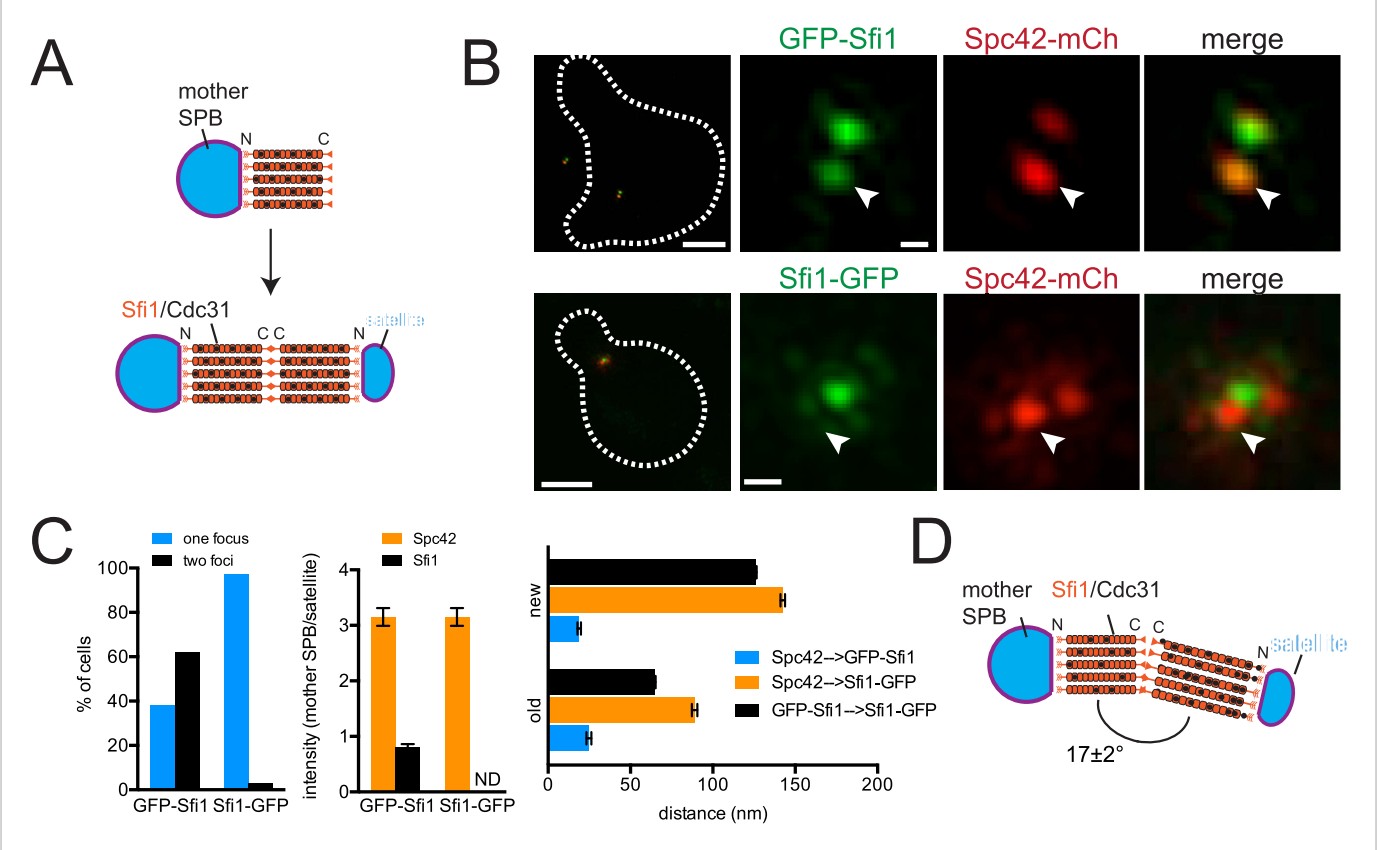

**Figure 2**. Structure of the half-bridge. (**A**) Top-down view of half-bridge showing the mother SPB and the Sfi1-Cdc31 filament extending in a polar fashion along the half-bridge; C-terminal end-to-end association forms an N-terminal end for satellite assembly. (**B**) Cells containing Spc42-mCherry and GFP-Sfi1 (SLJ9741) or Sfi1-GFP (SLJ10040) were α-factor arrested and imaged by SIM. On the left is a merged image showing the cell outline (dashes). Bar, 2 μm. Single channel and merged images of the mother SPB (arrowhead) and satellite. Bar, 200 nm. (**C**) Cells from **B** were quantitated and the percentage of cells containing two Sfi1 foci and the ratio of intensity between the mother SPB/mother proximal signal and satellite/distal signal is shown. ND, not determined. Distance was determined in three dimensions between Spc42-mCherry and GFP-Sfi1 or Spc42-mCherry and Sfi1-GFP on the old and new Sfi1 filament that is proximal and distal to the mother SPB, respectively. The distance between Sfi1 foci was calculated using data in *Figure 3E*. Error bars, standard error of the mean (SEM). (**D**) Modified schematic of bridge from **A**, showing Sfi1 and Cdc31 asymmetry and the bend detected by SIM.

(*Figure 2B*, merged panels) and the average distance between Spc42-mCherry at the mother SPB and Sfi1-GFP in the 'center' of the bridge is significantly shorter than the distance between Sfi1-GFP and the satellite (*Figure 2C*). Based on these observations, we hypothesize that newly formed Sfi1 is distinct from Sfi1 existing on the half-bridge from the previous cell cycle and that the new Sfi1 filament is built at an angle relative to the existing Sfi1 (*Figure 2D*). These asymmetries may be important for bridge expansion, contraction, and bending, which have previously been proposed to facilitate formation and NE insertion of the new SPB (*Adams and Kilmartin, 1999*, *2000*).

## Distribution of half-bridge components in the extended bridge using SPA-SIM

To compare the positions of Sfi1 N- and C-termini along with other half-bridge components, we developed computational methods to align dual color SIM images based on Spc42 fluorescence at the mother SPB and the satellite (see 'Materials and methods'). Conceptually similar to single particle analysis used in many cryoEM studies, SPA-SIM involves the generation of probability profiles depicting the average position of the second SPB component after alignment of many images using Spc42. This type of analysis incorporating a common fiducial marker is advantageous because it shows the likelihood that a protein is present in a given location based on many cells and allows for positional comparison between different proteins.

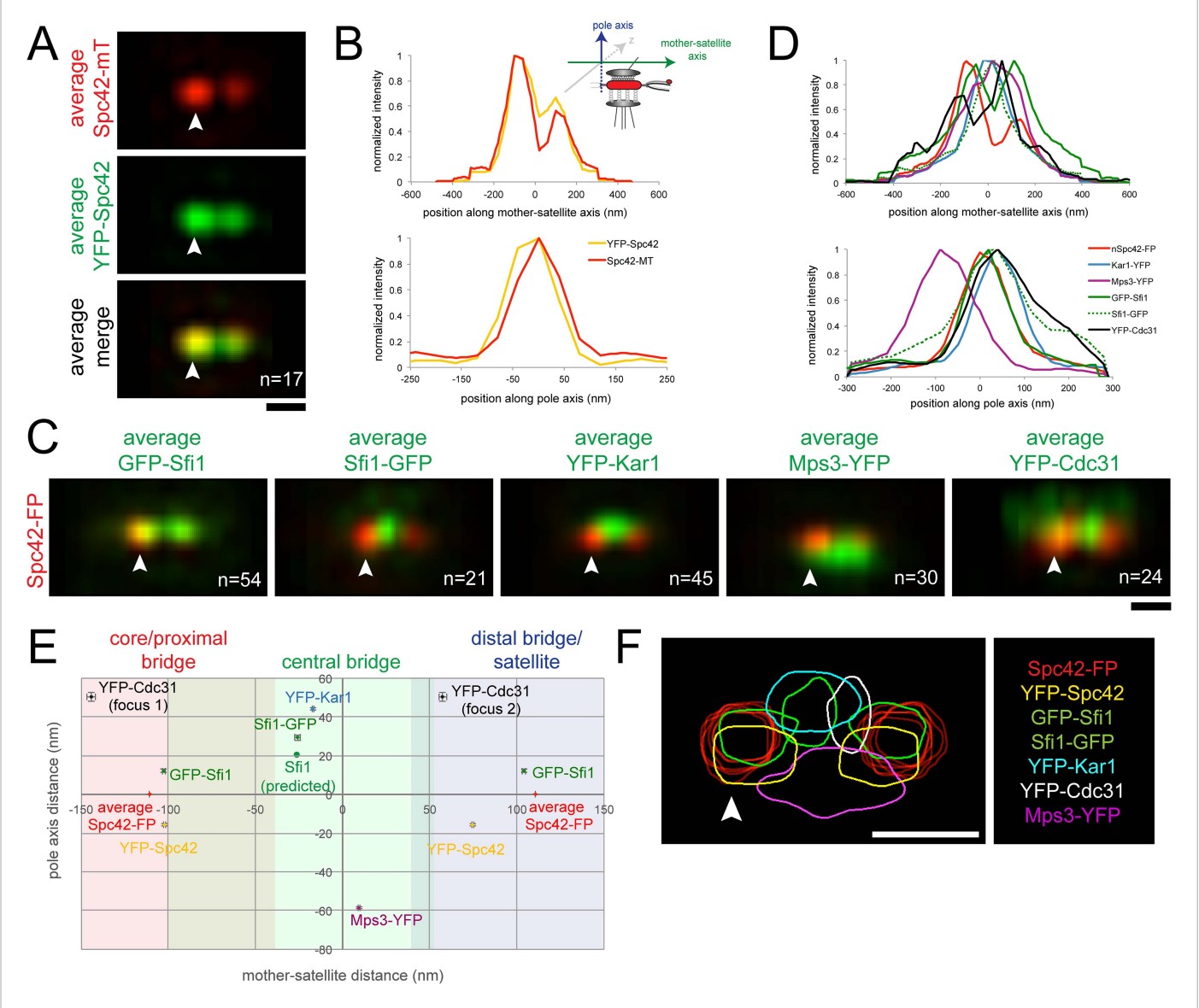

**Figure 3**. SPA-SIM analysis of the extended half-bridge. (**A**, **B**) YFP-Spc42-mTurquoise2 (SLJ9442) cells were arrested in α-factor and imaged by SIM. 17 images of the SPB (arrowhead) and satellite were aligned based on mTurquoise2 fluorescence to create the projection view shown in **A**. Bar, 200 nm. In **B**, graphs show the relative fluorescence intensity along the mother-satellite (top) and pole (bottom) axis, in nm, as depicted in the schematic. (**C**, **D**) SIM images of cells from *Figure 2B* and α-factor arrested YFP-Cdc31 (SLJ10084; representative image shown in *Figure 3—figure supplement 1*), YFP-Kar1 (SLJ9670) or Mps3-YFP (SLJ9454) strains containing Spc42-mTurquoise2 were aligned and projected. In **C**, Spc42-mTurquoise2 or Spc42-mCherry (red, denoted Spc42-FP) mark the mother SPB (arrowhead) and satellite and localization of the indicated half-bridge component is shown in green. N is indicated. (**D**) Normalized fluorescence intensity of each protein along the mother-satellite and pole axis is plotted. (**E**) To compare positional information between samples, the maximum intensity of GFP-Sfi1, Sfi1-GFP, YFP-Cdc31, YFP-Kar1, Mps3-YFP, and YFP-Spc42 distributions was determined in both axes and plotted using the center position between Spc42-mTurquoise2/mCherry at the mother SPB and satellite as the zero reference position. Localization of Kar1 and Mps3 to opposite sides of the bridge was confirmed by SIM, as shown in *Figure 3—figure supplement 1*. Error bars, SEM. Based on the full-width half-maximum (FWHM) values of Spc42 at the mother (110 nm, −180 to −30 nm), satellite (110 nm; 36 to 184 nm) and Sfi1-GFP (−25.4 nm; −102.4 to 51.6 nm) (*Table 1*), the bridge was divided into core/proximal, central and distal/satellite regions. The predicted position of Sfi1-GFP (Sfi1-predicted) based on the natural curvature of the NE of a nucleus of 1 μm is shown. (**F**) Contour map showing the distribution of fluorescent intensity at the extended half-bridge based on all images used in **A** and **C**. The C-terminally tagged Spc42 (Spc42-FP) in each sample is shown in red and other proteins are colored as indicated. Bar, 200 nm.

The following figure supplement is available for figure 3:

**Figure supplement 1**. Mps3 and Kar1 localize to opposite faces of the half-bridge.

As a proof-of-principle, we first aligned seventeen images from α-factor arrested cells containing a version of Spc42 fused at its N-terminus to YFP and at its C-terminus to mTurquoise2 (YFP-Spc42-mTurquoise2). Based on EM, the C-terminus of Spc42 is above the NE in intermediate layer 2 of the SPB, which is located between the central and outer plaque; the Spc42 N-terminus is located in the central plaque, which is spaced 10.8 nm from intermediate layer 2 towards the NE (*Bullitt et al., 1997*; *O'Toole et al., 1999*; *Muller et al., 2005*). Because two reference points were needed for positioning, we selected C-terminal-labeled Spc42 at the mother SPB and satellite in our raw SIM images based on differences in intensity using the mTurquoise2 signal. We fit these manually selected positions to three-dimensional Gaussian functions in order to determine their centers with greater accuracy. Images were then realigned in three dimensions so that the centers of Gaussian fits were arranged along the x axis with the center point between the mother and satellite at zero. Therefore, in aligned images, the x axis represents the mother-satellite axis while the y axis represents the pole axis as depicted in *Figure 3B*. Examination of YFP intensity in aligned images showed a high degree of correlation in the mother-satellite axis, while in the pole axis the ~15 nm shift in intensity maxima is consistent with the position of the N-terminus in the central plaque and the C-terminus in intermediate layer 2 (*Figure 3A,B*; *Table 1*). These data demonstrate that we have excellent color alignment in our raw images and that the use of Spc42 enables comparison of positional information between multiple images with high confidence using our alignment methods.

Alignment of GFP-Sfi1 and Sfi1-GFP images using Spc42-mCherry further demonstrated the validity of this approach and provided additional insights into its organization at the extended half-bridge that were not ascertained by inspection of individual images or by high resolution EM, PALM and stochastic optical reconstruction microscopy (STORM) analysis (*Kilmartin, 2003*; *Li et al., 2006*; *Seybold et al., 2015*). Not only is Sfi1-GFP displaced, but GFP-Sfi1 is also cytoplasmically shifted relative to Spc42 (*Figure 3C–E*; *Table 1*). These shifts of 29.4 ± 1.4 nm (Sfi1-GFP) and 12.1 ± 1.0 nm (GFP-Sfi1) are significantly less than the ~100 nm resolution achievable by SIM alone (*Gustafsson et al., 2008*), suggesting that our SPA-SIM approach can provide at least a 5–10-fold improvement in intensity distribution localization. A 17 ± 2° bend in the extended half-bridge was readily apparent as was the asymmetry between old and new Sfi1 filaments and the increased intensity of GFP-Sfi1 at the satellite

**Table 1**. Probability distribution fit parameters from *Figure 3*

| Sample | Query | | | | Spc42-reference | | |
|---|---|---|---|---|---|---|---|
| | x | y | FWHM$_x$ | FWHM$_y$ | x | FWHM$_x$ | FWHM$_y$ |
| YFP-Spc42 | −96.4 (2.2) | −15.6 (0.6) | 141.1 (2.7) | 108.1 (1.4) | −104.9 (1.3) | 140.2 (3.0) | 104.6 (1.2) |
| | 80.4 (2.8) | −15.6 (0.6) | 148.4 (5.0) | | 104.9 (1.3) | 136.4 (5.5) | |
| Kar1-YFP | −12.4 (2.8) | 44.2 (0.6) | 178.8 (3.3) | 118.8 (1.1) | −106.0 (1.6) | 136.8 (4.6) | 112.9 (1.4) |
| | | | | | 106.0 (1.6) | 129.8 (8.0) | |
| Mps3-YFP | 10.9 (2.3) | −58.5 (0.5) | 318.1 (17.8) | 158.1 (1.2) | −109.2 (1.3) | 158.0 (3.7) | 114.7 (1.3) |
| | | | | | 109.2 (1.3) | 133.7 (6.5) | |
| GFP-Sfi1 | −100.3 (7.5) | 12.1 (0.8) | 153.6 (3.8) | 110.3 (2.2) | −108.4 (1.5) | 139.5 (3.2) | 131.3 (1.3) |
| | 106.4 (8.3) | 12.1 (0.8) | 194.9 (14.5) | | 108.4 (1.5) | 131.4 (6.1) | |
| Sfi1-GFP | −25.9 (3.2) | 29.4 (1.5) | 214.3 (6.6) | 153.9 (5.1) | −110.8 (2.0) | 147.0 (3.4) | 123.8 (1.3) |
| | | | | | 110.8 (2.0) | 140.1 (8.7) | |
| YFP-Cdc31 | −152.4 (15.4) | 50.4 (2.3) | 247.9 (29.4) | 180.7 (7.6) | −118.9 (1.4) | 195.7 (3.7) | 158.2 (1.1) |
| | 49.2 (6.9) | 50.4 (2.3) | 143.8 (14.1) | | 118.9 (1.4) | 156.7 (4.8) | |

Notes: All values are in nm. In all cases 'x' refers to the mother-satellite axis and 'y' refers to the pole axis. The zero x and y axis value is defined as midway between the C-terminal Spc42 reference peaks in our realignment scheme. The full-width half-maximum (FWHM) is 2.35 times the standard deviation of the Gaussian fit, and this can be converted into 95% integral values by multiplying each by 1.7. In all cases the mother spindle pole body (SPB) peak is shown on the first line and the satellite peak, if applicable, is shown on the second line. FWHM$_y$ in cells containing two foci was determined by the averages over the two peaks. Errors are in parentheses and are standard deviations from Monte Carlo random fits as described in 'Materials and methods'.

(*Figures 2D, 3E*). This kink in the extended half-bridge occurs in the direction of the pole axis, which is perpendicular to the NE. It is greater than the natural bend predicted based on the curvature of the NE, which would be ~7° based on a 1 μm nuclear diameter (*Figure 3E*).

Unlike Sfi1, which appeared to form a filament with a distinct polarity, Mps3 and Kar1 molecules were distributed along the half-bridge in asynchronous and α-factor arrested cells (*Figure 3—figure supplement 1*; data not shown) (*Seybold et al., 2015*). Probability profiles of Mps3-YFP and YFP-Kar1 showed that both proteins are displaced along the pole axis (*Figure 3C,D*; *Table 1*). Co-localization of Mps3-mTurquoise2 and YFP-Kar1 showed that the proteins are located on opposite sides of the extended bridge (*Figure 3—figure supplement 1*). Previous immunoEM analysis showed that Mps3 is primarily at the inner NE face of the half-bridge and Kar1 is primarily cytoplasmic (*Vallen et al., 1992b*; *Jaspersen et al., 2002*), so we have shown distributions according to their proper orientation (*Figure 3C–F*). A recent half-bridge model suggests that Kar1 concentrates at a region of extended half-bridge located near the Sfi1 C-terminus (*Seybold et al., 2015*). Based on the full-width half-maximum (FWHM) values of our Gaussian fits, YFP-Kar1 is more broadly distributed along the mother-satellite axis than Sfi1-GFP (179 ± 3 nm compared to 154 ± 4 nm; *Table 1*), which was confirmed by inspection of the probability profiles and individual SIM images (*Figure 3C,F*; data not shown). The broader distribution that we observe for Kar1 along the extended half-bridge may be important for its association with pore components of the SPB (discussed below).

N- or C-terminal fusions of fluorescent proteins to *CDC31* were lethal when expressed under the endogenous promoter; however a *cdc31Δ::MET-YFP-CDC31* strain was viable and YFP-Cdc31 was detected as one or two foci in most α-factor arrested cells (*Figure 3—figure supplement 1*). Alignment of images using Spc42-mCherry present in the strain showed that the brighter focus (focus 2) was located at a distal region of the bridge interior to the N-terminus of Sfi1 (*Figure 3C–E*; *Table 1*). Detection of YFP-Cdc31 in foci, as opposed to distributed along the bridge like YFP-Kar1 and Mps3-YFP, was unexpected based on its genetic and physical interactions with Kar1 and structural studies in vitro showing that Cdc31 binds along the tryptophan-containing repeats in the central region of Sfi1 (*Biggins and Rose, 1994*; *Vallen et al., 1994*; *Spang et al., 1995*; *Li et al., 2006*). While it is possible that YFP-Cdc31 does not accurately report the localization of native Cdc31, it is the only copy of Cdc31 in these cells and therefore is sufficient to carry out the essential function(s) of Cdc31 in SPB duplication. The unequal distribution of YFP-Cdc31 on the bridge may be related to Sfi1, which also showed increased levels and possibly a different structure (based on its length) in the distal region of the extended bridge (*Figure 2B,D*, *Figure 3C–F*; *Table 1*).

## Half-bridge elongation is a distinct step that precedes satellite assembly

Our ability to observe two GFP-Sfi1 foci allowed us to ask when in the cell cycle SPB duplication initiates. During most of the cell cycle, GFP-Sfi1 appeared as a single focus; however, two foci were detected at one or both poles at the end of anaphase in many (63%, 25/40) large budded cells and at the pole in nearly all early G1 cells (90%, 26/29) (*Figure 4A*). In these same cells, two foci of Spc42-mCherry were never seen at the SPB in anaphase cells and only 31% of the poles in early G1 cells had two Spc42 foci. The late mitotic timing of half-bridge elongation is consistent with recent studies showing that dephosphorylation of Cdk1 sites in Sfi1 by the Cdc14 phosphatase licenses a new round of SPB duplication (*Avena et al., 2014*; *Elserafy et al., 2014*). Our observation of an SPB duplication structure in mitosis is earlier than described by EM, perhaps because it is a transient step that is only observed in a small fraction of cells (*Byers and Goetsch, 1974*, *1975*). In both late mitotic and early G1 cells, an extended half-bridge was detected before the Spc42-containing satellite was observed (*Figure 4A*), suggesting that bridge elongation is a distinct step in SPB duplication that must occur prior to formation of the satellite.

Previous EM analysis of mutant alleles in the Mps1 kinase showed cells containing *mps1-1* arrest with an elongated bridge lacking a satellite, which is reminiscent of certain *SFI1* mutants (*Winey et al., 1991*; *Kilmartin, 2003*; *Anderson et al., 2007*). Thus, it seems likely that Mps1 regulates progression from the elongated bridge state we observed in cycling cells to the satellite-bearing stage in SPB duplication. To test this idea, we examined the localization of GFP-Sfi1 in *mps1-1* cells by both immunoEM and SIM. Three pools of GFP-Sfi1 were found in the extended half-bridge of *mps1-1* mutant cells by immunoEM: 5/19 gold particles were proximal to the mother SPB and 10/19 gold particles were distal. The remaining 4/19 particles were located closer to the center of the bridge

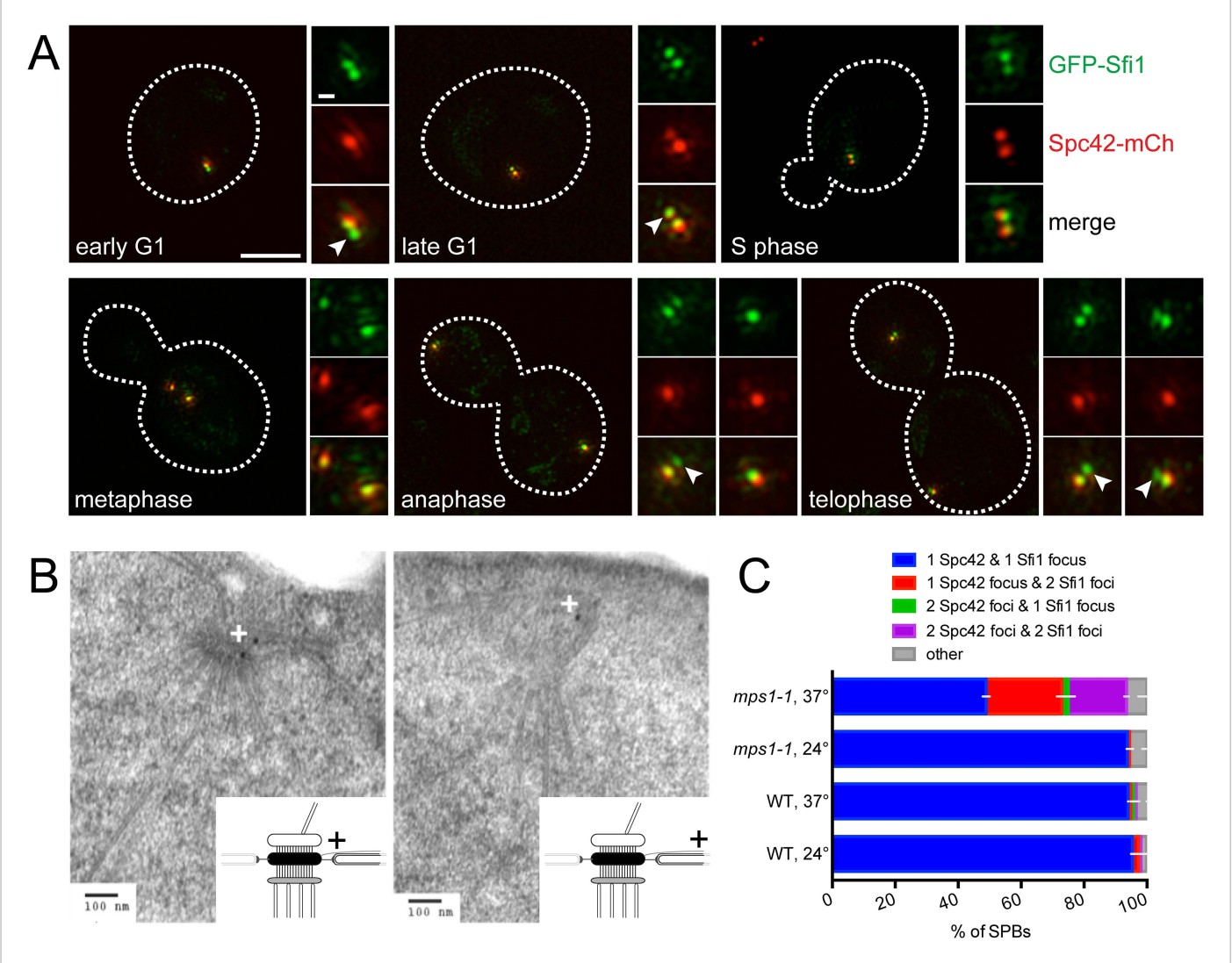

**Figure 4**. Half-bridge elongation is a discrete step in SPB duplication. (**A**) SIM images from asynchronously growing cells (SLJ9741) containing GFP-Sfi1 (green) and Spc42-mCherry (red). A merged image showing the cell outline (dashes) was used together with spindle length to approximate the cell cycle position indicated. Bar, 2 μm. The SPB(s) are shown to the right of each cell. Arrowheads in the merged images point to the satellite. Bar, 200 nm. (**B**) GFP-SFI1 mps1-1 cells (SHJ3829) were grown to log phase at 24°C, shifted to restrictive temperature (37°C, 4 hr) and then prepared for immunoEM with nanogold secondary label. Shown are two representative cells with labeling at the SPB (left) and the distal tip of the elongated half-bridge (right). Some cells also contained labeling closer to the center of the elongated half-bridge (left). +: label representation. (**C**) Wild-type (JA372) and mps1-1 (JA368) cells containing GFP-Sfi1 and Spc42-mCherry were grown at 24°C or shifted to 37°C for 4 hr then analyzed by SIM. Because mps1-1 cells arrest in mitosis at the non-permissive temperature (**Winey et al., 1991**), only large budded cells were examined. The SPBs from early mitotic wild-type cells showed co-localization of GFP-Sfi1 and Spc42-mCherry (95 ± 4%, n = 90, at 24°C or 94 ± 1%, n = 117, at 37°C), with 87 ± 14% (24°C) and 80 ± 17% (37°C) displaying co-localization at both poles. 94 ± 2% (n = 104) of mitotic mps1-1 cells grown at 24°C showed the same localization as wild-type, with 86 ± 1% of cells exhibiting co-localization at both poles. At 37°C, 49 ± 7% of cells showed co-localization of GFP-Sfi1 and Spc42-mCherry, with the majority of cells displaying a single focus of each (40% of all mps1-1 cells at 37°C). 24 ± 7% and 19 ± 3% of SPBs (n = 85) contained a single focus of Spc42-mCherry or two Spc42-mCherry, respectively, with two GFP-Sfi1 foci. Error bars, SEM.

(**Figure 4B**). SIM of wild-type cells at 24°C or 37°C or mps1-1 mutants grown at 24°C showed that virtually all the early mitotic SPBs in large budded cells contained a single focus of both Spc42-mCherry and GFP-Sfi1. In mps1-1 mutants shifted to 37°C, the frequency of this class of SPBs decreased and SPBs containing two GFP-Sfi1 foci increased, including SPBs with two Spc42-mCherry foci (representing SPBs containing an elongated half-bridge and a satellite/duplication plaque or duplicated side-by-side SPBs) or one Spc42-mCherry focus (representing a SPB with an elongated

bridge lacking a satellite) (*Figure 4C*). The mixed phenotype is consistent with molecular and genetic analysis of Mps1 showing that Mps1 function is required at multiple points in SPB duplication (*Schutz and Winey, 1998*; *Jones et al., 2005*; *Araki et al., 2010*). Collectively, these data demonstrate that half-bridge elongation is a distinct step of SPB duplication that begins in late mitosis and that the conversion from an elongated half-bridge to a satellite-bearing SPB requires the function of Mps1.

## Spc29 and Spc42 are the major satellite components

In addition to Spc42, the satellite is predicted by immunoEM to contain three additional components of the core SPB: Spc29, Nud1, and Cnm67 (*Adams and Kilmartin, 1999*). To test if these localized to the satellite by SIM, we arrested cells at the satellite-bearing stage of SPB duplication using α-factor for 3 hr. The duration of α-factor treatment ensured that more than 95% of cells arrested in G1; however, since most yeast strains have an ~90 min generation time at 30°C, many cells were likely in G1 for an extended period of time compared with the time normal cycling cells spend in this cell cycle stage.

As shown in *Figure 5A,B*, Spc42-mTurquoise2 and Spc29-mTurquoise2 were observed by SIM at the mother SPB (defined by Spc110-YFP) and in the satellite in most cells (89% [n = 38] and 74% [n = 33], respectively) treated with mating pheromone. The fluorescence intensity ratio at the mother SPB vs the satellite was lower for Spc42-mTurquoise2 (1.9 ± 0.2) compared to Spc29-mTurquoise2 (2.8 ± 0.5), suggesting that more Spc42 is present in the satellite relative to Spc29 and/or more Spc29 is at the mother SPB relative to Spc42 (*Figure 5B*).

Nud1-mTurquoise2 was observed at the satellite in many (67%, n = 31) but not all α-factor arrested cells (*Figure 5A,B*). The intensity of Nud1-mTurquoise2 at the mother SPB relative to the satellite (1.9 ± 0.3) was equivalent to that of Spc42-mTurquoise2 so it is unlikely that an inability to detect Nud1-mTurquoise2 at the satellite was the cause of the reduced number of satellites observed (*Figure 5B*). One possibility is that Nud1 is a more peripheral satellite component that may assemble later as the satellite matures into a duplication plaque (see below). The fact that the distance between the mother SPB and satellite is greater when measured by Nud1-mTurquoise2 (275 ± 12 nm) than Spc29-mTurquoise2 (228 ± 13 nm) or Spc42-mTurquoise2 (225 ± 10 nm) (*Figure 5B*) would be consistent with this idea, particularly if the bridge adopts a bent shape (see *Figures 2D, 3E,F*).

Cnm67 showed high variability at the satellite (ranging from 0–43%), even using five different strains and three distinct fluorescent protein tags (*Figure 5—figure supplement 1*). This variability has been observed previously with anti-Cnm67 antibodies using immunoEM (*Adams and Kilmartin, 1999*). The high average ratio between mother SPB and satellite for Cnm67 (2.9 ± 0.2) and its presence at the satellite in an average of 37% of cells suggests that Cnm67 is not essential for satellite assembly and it is present in lower amounts compared with Spc42 and Nud1, which was confirmed by visual inspection of images (*Figure 5—figure supplement 1*; data not shown). Further analysis of Cnm67 was omitted due to difficulties in reproducibly observing it at the satellite.

## The satellite is assembled in a stepwise manner

To effectively examine the timing at which individual components assemble into the satellite, we started with a highly synchronized mitotic cell population by employing a metaphase arrest/release protocol involving depletion of the anaphase activator Cdc20 (*Figure 5—figure supplement 2A*). Because there are currently no known cytological markers for the early steps of SPB duplication other than structures visible by EM, the synchronization ensured we compared equivalent duplication intermediates. Satellite component assembly was monitored using fluorescently tagged proteins by SIM in cells released from metaphase.

In cells containing Spc42-mTurquoise2 and Nud1-YFP, we observed two foci of Spc42-mTurquoise2 as early as 45 min following release from metaphase, at about the same time a very small bud was detected (*Figure 5C,D* and *Figure 5—figure supplement 2B*). Of the very small budded cells at this time point, 82% (77/94) showed two foci of Spc42-mTurquoise2, which is indicative of satellite assembly. Most of the cells with two Spc42-mTurquoise2 foci (90%, 69/77) had a single focus of Nud1-YFP that was coincident with the more intense focus of Spc42-mTurquoise2, which is consistent with our theory that Nud1 assembles into the satellite later than Spc42. We observed Nud1-YFP at the SPB in the majority of SPBs at later time points (83% at 60 min, 48/58), as illustrated in *Figure 5C* (60 min)—both cells have two Spc42-mTurquoise2 foci, and the cell with the larger bud that has progressed

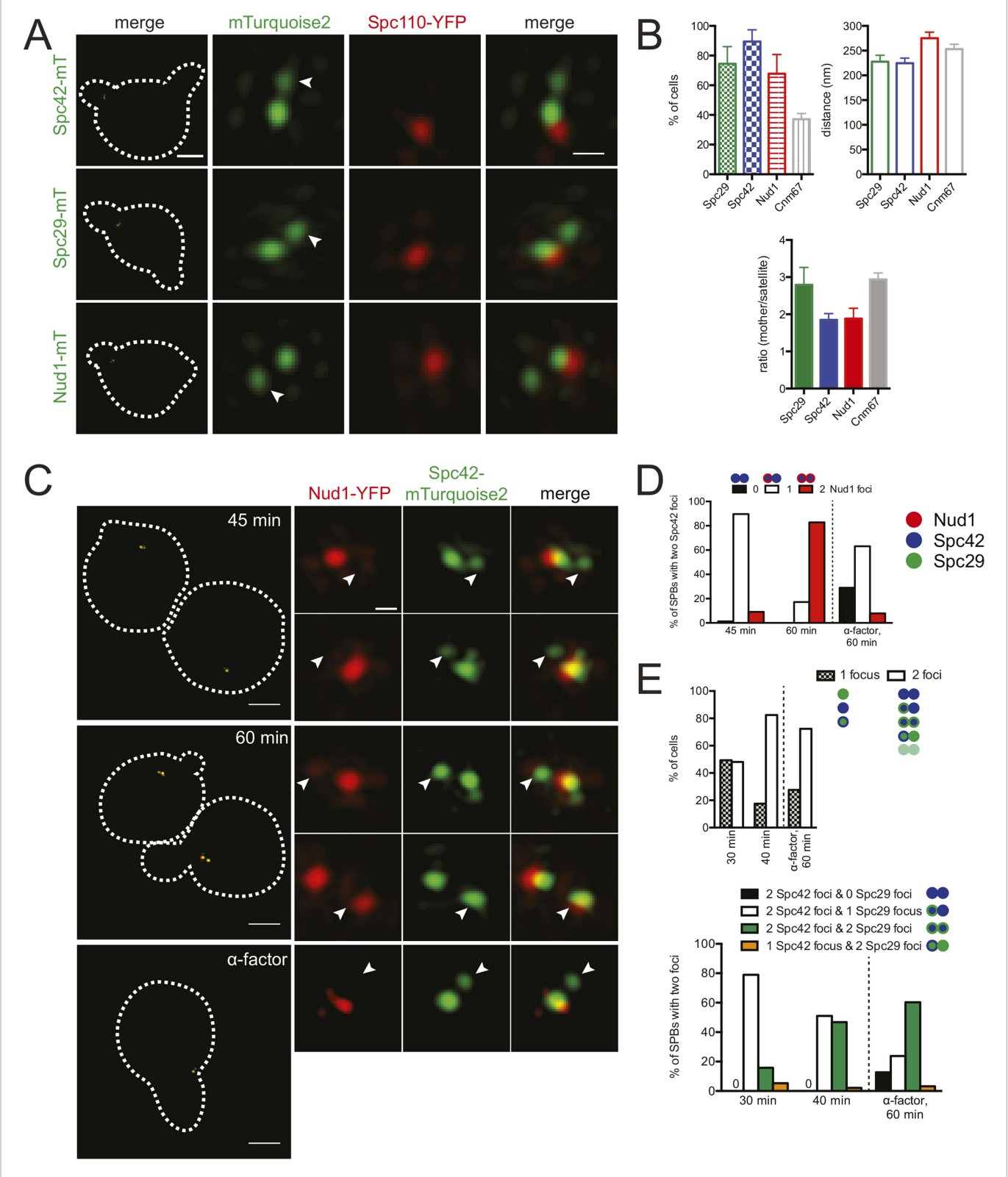

**Figure 5**. Temporal control of satellite assembly. (**A**, **B**) Cells containing Spc110-YFP (red) and Spc42-mTurquoise2 (SLJ8980), Spc29-mTurquoise2 (SLJ8820), or Nud1-mTurquoise2 (SLJ9099) (green) were α-factor arrested for 3 hr and imaged using SIM. The cell is shown on the left with dashes indicating the cell boundary. Bar, 2 μm. Single channel and merged images of the mother SPB and satellite (arrowhead). Bar, 200 nm. (**B**) Cells from **A** were

*Figure 5. continued on next page*

*Figure 5. Continued*

quantitated and the percentage of α-factor arrested cells containing two foci (mother and satellite) is shown, along with the distance from mother to satellite and the ratio of their intensity. The mother SPB overlaps or is adjacent to Spc110-YFP. Data for Cnm67 is an average based on data shown in *Figure 5—figure supplement 1*. Errors, SEM. (**C**, **D**) A metaphase arrested *MET-CDC20* strain (SLJ9720) containing Spc42-mTurquoise2 (green) and Nud1-YFP (red) was released into the cell cycle using SC-methionine media. Aliquots were taken every 15 min to determine budding index and for analysis of satellite assembly by SIM. (**C**) Images from the 45 and 60 min time points are shown together with cells released into α-factor for 60 min. The cells are shown on the left with dashes indicating the cell boundary. Bar, 2 µm. Single channel and merged images of the SPB and the satellite (arrowhead). Bar, 200 nm. (**D**) Percentage of cells with SPBs that have two foci of Spc42-mTurquoise2 (the pole and the satellite) and 0 (black bar), 1 (white bar) or 2 (red bar) foci of Nud1 is plotted for each time point. The reciprocal plot as well as an experiment in which Nud1 is labeled with mTurquoise2 and Spc42 with YFP is in *Figure 5—figure supplement 2B,C*. A red and blue dot represent Nud1 and Spc42, respectively. (**E**) As in **C**, a *MET-CDC20* strain (SLJ10009) containing Spc29-YFP and Spc42-mTurquoise2 was released and samples were collected every 10 min. The percentage of cells at 30 and 40 min after release from metaphase and the sample released into α-factor for 60 min with a single focus or two foci of either Spc42-mTurquoise2 and/or Spc29-YFP at the SPB is plotted. A blue and green dot represent Spc42 and Spc29, respectively. At 30 min, approximately 50% of cells have a single unduplicated SPB at the pole but this decreased to 10% by 40 min. Note that none of the cells at 40 min had two foci of just Spc29-YFP. All other combinations were observed and the percentages are shown in the graph below. Reciprocal plots are in *Figure 5—figure supplement 2D*.

The following figure supplements are available for figure 5:

**Figure supplement 1**. Cnm67 at the satellite.

**Figure supplement 2**. Temporal control of satellite assembly.

further in the cell cycle contains Nud1-YFP at both poles. As a control, we also released cells from metaphase into α-factor. While we observed two foci of Spc42-mTurquoise2 in most (60%, 38/63) cells 60 min following release from metaphase, only 8% (5/63) had two foci of Nud1-YFP (*Figure 5C,D* and *Figure 5—figure supplement 2B*). The reduction in the fraction of α-factor-containing SPBs with Nud1 at the satellite in this experiment compared with asynchronous cells arrested in α-factor for 3 hr (*Figure 5A,B*) was also observed in Spc42-YFP/Nud1-mTurquoise2 cells (*Figure 5—figure supplement 2C*; 25%, 14/55). If cells were released into α-factor for 3 hr, the fraction of cells containing two foci of Spc42-YFP and Nud1-mTurquoise2 rose to 71% (46/64), showing that Nud1 accumulates at the satellite during a prolonged pheromone arrest (*Figure 5—figure supplement 2C*).

When Spc42-mTurquoise2/Spc29-YFP cells were synchronized and released as described above, we observed a satellite that contained Spc42-mTurquoise2, Spc29-YFP, or both in 48% (n = 79) of cells 30 min following release from metaphase (*Figure 5E* and *Figure 5—figure supplement 2D*). Of these SPBs, 30/38 have two foci of Spc42-mTurquoise2 and a single Spc29-YFP focus, 2/38 have two foci of Spc29-YFP and a single Spc42-mTurquoise2, and 6/38 have two closely spaced foci of both proteins, similar to what was observed in cells released into α-factor. This suggests that Spc42 precedes Spc29 in recruitment to the satellite; however, we cannot exclude the possibility this is due to detection issues since we were unable to create a strain containing *MET-CDC20/SPC42-YFP/SPC29-mTurquoise2*. However, the observation that the percentage of SPBs containing two Spc29-YFP foci in addition to two foci of Spc42-mTurquoise2 increased from 15% at 30 min to 47% at 40 min supports that Spc42 assembles prior to Spc29 (*Figure 5E* and *Figure 5—figure supplement 2D*).

During metaphase arrest induced by the temperature-sensitive *cdc20-1* mutant, SPB size is over twice the size as in wild-type cells in mitosis (*O'Toole et al., 1997*). To exclude the possibility that accumulation of SPB components during our synchronization protocol using *MET-CDC20* affected the order of satellite assembly, we examined asynchronously growing cells. Because SPB duplication occurs during G1 phase (*Byers and Goetsch, 1974*), we focused on unbudded cells, finding that 25–50% of cells contained a single SPB labeled with a focus of Nud1-mTurquoise2/Spc42-YFP (48%, n = 114), Spc42-mTurquoise2/Nud1-YFP (41%, n = 144), Spc42-mTurquoise2/Spc29-YFP (45%, n = 134), or Spc29-CFP/YFP-Spc42 (25%, n = 77) (*Figure 5—figure supplement 2E*). In the remaining cells, two closely spaced foci were observed. In agreement with our results described above, most contained two foci of Spc42 and one or two foci of Nud1 or Spc29 (*Figure 5—figure supplement 2E*).

Thus, the temporal order of SPB assembly appears to initiate with bridge elongation by the addition of Sfi1, followed by Spc42 deposition followed rapidly by Spc29 and Nud1 assembly later. Interestingly, overexpression of Spc42 leads to a lateral expansion of the central layer of the core SPB

into a structure sometimes referred to as the superplaque (*Donaldson and Kilmartin, 1996*; *Castillo et al., 2002*). The ability of Spc42 overproduction alone to form this structure and its regulation by Mps1 are consistent with our data showing that Spc42 is the initial protein deposited at the satellite and that satellite formation is Mps1 dependent.

## Localization of other SPB components

We conducted a survey of the other ten SPB components by SIM to look at their different locations within the core SPB, and in some cases distinct locations in cycling cells vs cells undergoing SPB duplication (*Figure 1A*). As depicted in *Figure 6—figure supplement 1*, this was observed for Spc72-YFP and Tub4-mTurquoise2; both localize to the extended half-bridge in cells treated with α-factor, consistent with previous studies showing microtubule nucleation off the bridge during G1 and during mating (*Byers and Goetsch, 1974*, *1975*; *Pereira et al., 1999*). Molecular and cytological studies showed that the N- and C-termini of Spc110 are located at the inner and central plaque, respectively (*Geiser et al., 1993*; *Kilmartin et al., 1993*; *Kilmartin and Goh, 1996*; *Spang et al., 1996*; *Sundberg et al., 1996*). We found that the C-terminus of Spc110 co-localized with calmodulin (Cmd1) while the N-terminus of Spc110 overlapped with one of the two foci seen in strains containing γ-tubulin complex components Spc97-mTurquoise2 or Spc98-mTurquoise2 (*Figure 6—figure supplement 1*).

To visualize the SPB pore (the protein structure that surrounds the core SPB and anchors the SPB in the NE), we exploited the fact that SPB diameter, and thus pore diameter, increases with ploidy (*Byers and Goetsch, 1974*; *Bullitt et al., 1997*). Localization of the membrane proteins Mps2-mTurquoise2 and Ndc1-YFP in tetraploid cells revealed a ring-like structure in some cells and linear structures in others (*Figure 6A*). Ndc1-YFP fluorescence was also observed in puncta throughout the NE, presumably at nuclear pore complexes (*Chial et al., 1998*). Ring-like structures at the SPB could be visualized in haploid and diploid cells, although the smaller pore diameter made the central region of the ring more challenging to resolve (*Figure 6A*; data not shown). Based on the fact that Mps2 and Ndc1 are integral membrane proteins involved in SPB insertion into the NE (*Chial et al., 1998*; *Munoz-Centeno et al., 1999*), the donut shaped rings are likely SPBs in transverse section with Mps2 and Ndc1 localizing to the pore membrane around the SPB. The linear structures are the SPB in longitudinal section.

A complex set of genetic and physical interactions suggests that Mps2 and Ndc1 anchor the SPB in the NE via binding to the soluble proteins Bbp1 and Nbp1, respectively, which associate with components of the core SPB such as Spc29 (*Schramm et al., 2000*; *Araki et al., 2006*; *Sezen et al., 2009*; *Casey et al., 2012*; *Chen et al., 2014*). Based on this data, we anticipated finding Nbp1 and Bbp1 in ring-like structures, similar to those seen with Ndc1 and Mps2. Therefore, it was surprising that Nbp1-YFP was not observed in ring-like structures at any time during the cell cycle. Instead, it localized as a discrete focus that co-localized with a region of the Ndc1-mTurquoise2 ring (*Figure 6A*). In most cells (93%, n=100), Bbp1 was also observed as one or two foci; however in 7% of cells, we saw ring-like structures of Bbp1-mTurquoise2 that co-localized with Ndc1-YFP (*Figure 6A*). The foci formed by Bbp1-mTurquoise2 were adjacent to the bridge (*Figure 6B*, top panels). In a fraction of cells with emerging buds, two foci of Bbp1-mTurquoise2 could be seen flanking Sfi1-YFP (*Figure 6B*, bottom panels), reminiscent of the localization pattern of satellite components. This suggests that Bbp1 and possibly other pore membrane components of the SPB may localize to early SPB duplication structures.

## Pore components localize to the satellite region early in SPB duplication

To test the idea that components of the pore membrane localize to the newly forming SPB, we examined the localization of Bbp1, Mps2, Ndc1, and Nbp1 fused to mTurquoise2 in α-factor arrested cells containing Spc110-YFP by SIM (*Figure 6C*). Under these conditions, Mps2-mTurquoise2 and Bbp1-mTurquoise2 were visible at both ends of the bridge in 62% (n = 21) and 63% (n = 47) of cells examined, respectively (*Figure 6C,D*). The distance between the two foci was approximately equal to the values for satellite components (*Figure 5B*). In some cells, Mps2-mTurquoise2 was not observed in two foci but rather was distributed across the region that is likely the bridge, possibly because the SPB is oriented in a longitudinal section (*Figure 6C*, arrowhead). Short linear structures were also observed in α-factor arrested cells containing Nbp1-mTurquoise2 and to a lesser extent Ndc1-mTurquoise2 (*Figure 6C*). Collectively, these data suggest that the Nbp1 and Ndc1 may localize to the extended half-bridge region, and Mps2 and Bbp1 accumulate on the mother SPB and satellite in α-factor arrested cells.

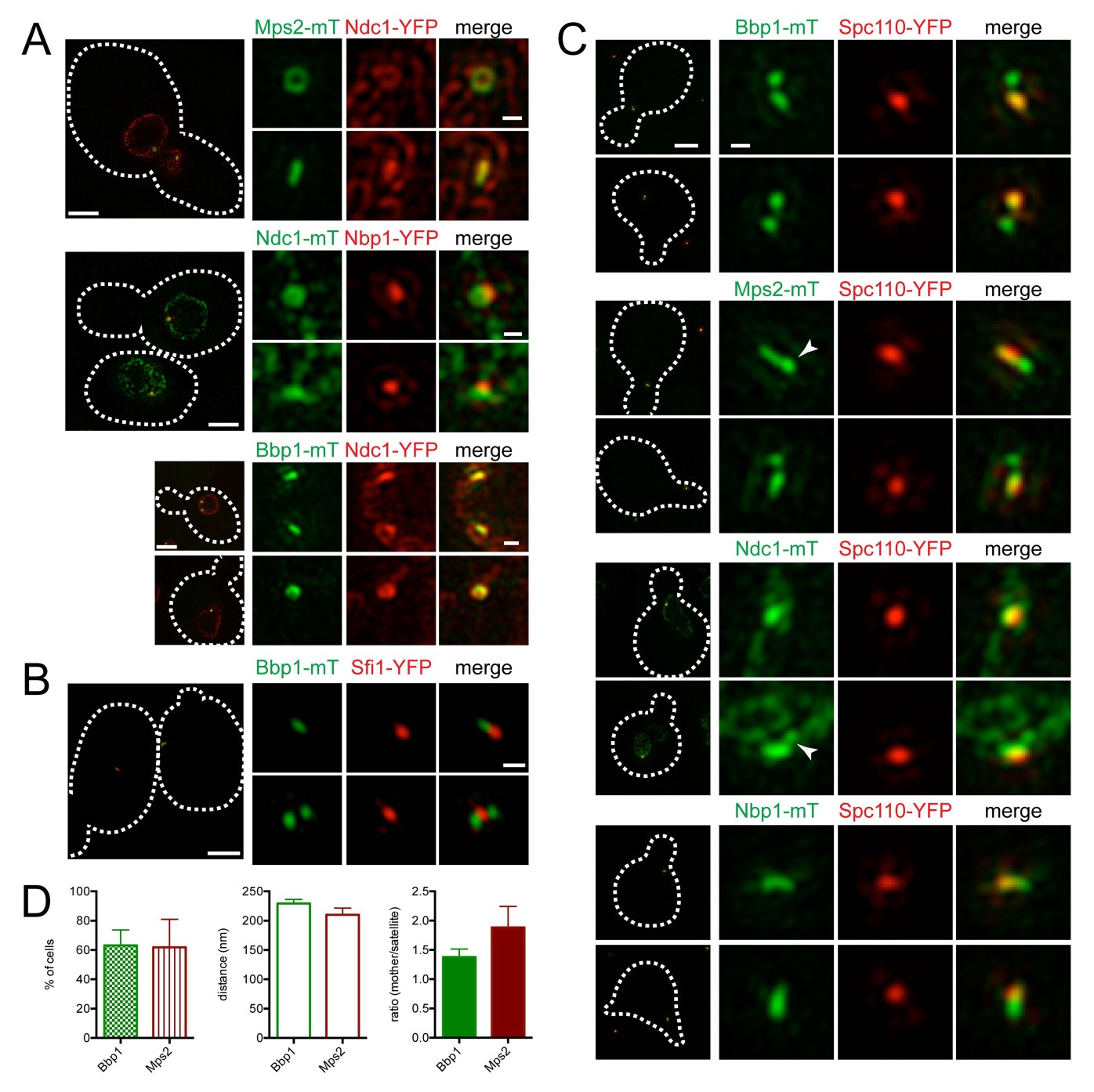

**Figure 6**. Localization of SPB pore components to the membrane region and half-bridge/satellite. (**A**) Asynchronously grown Mps2-mTurquoise2/Ndc1-YFP (SLJ8102), Ndc1-mTurquoise2/Nbp1-YFP (SLJ8263), and Bbp1-mTurquoise2/Ndc1-YFP (SLJ9231) were examined for evidence of a pore-like structure formed by Ndc1 and the other protein. (**B**) Location of Bbp1-mTurquoise2 (green) and Sfi1-YFP (red) in asynchronous (SLJ9035) cells. (**C, D**) Cells containing Spc110-YFP (red) and Bbp1-mTurquoise2 (SLJ10019), Mps2-mTurquoise2 (SLJ8084), Ndc1-mTurquoise2 (SLJ10018), or Nbp1-mTurquoise2 (green) were α-factor arrested and imaged using SIM. In **A–C**, the cell is shown on the left with dashes indicating the cell boundary. Bar, 2 μm. Single channel and merged images of the SPB region(s). Bar, 200 nm. In **C**, arrowheads point to the satellite region. (**D**) Bbp1 and Mps2 cells from **C** were quantitated and the percentage of α-factor arrested cells containing two foci (mother and satellite) is shown, along with the distance from mother to satellite and the ratio of their intensity. Error bars, SEM. The localization of other SPB components is shown in *Figure 6—figure supplement 1*.

The following figure supplement is available for figure 6:

**Figure supplement 1**. Localization of the γ-tubulin complex and linkers.

We used asynchronously dividing cells to examine the distribution of pore membrane proteins throughout the cell cycle (using bud size, SPB number, and distance between poles to approximate cell cycle position) to understand their distribution during SPB duplication and to provide insights into the temporal control of pore membrane assembly. As shown in *Figure 7A*, Bbp1-YFP was present in foci at the mother SPB and satellite in unbudded early G1 cells. Nbp1-mTurquoise2 initially localized to the mother SPB, then assembled in the region between Bbp1-YFP foci before forming two separate foci near the mother SPB and satellite as SPB duplication concluded in early S phase. The remnant of Nbp1-mTurquoise2 in the region between the two poles was no longer detected in medium budded cells, and both proteins localized to the SPB throughout the rest of the cell cycle. Similarly, Ndc1-YFP (*Figure 7B*) and Ndc1-mTurquoise2 (*Figure 7C*) also accumulated in the region between the mother SPB and the satellite in unbudded cells but later matured into two foci. These data strongly suggest that Bbp1 and Mps2 are recruited to the satellite early in its formation, ahead of Ndc1 and Nbp1.

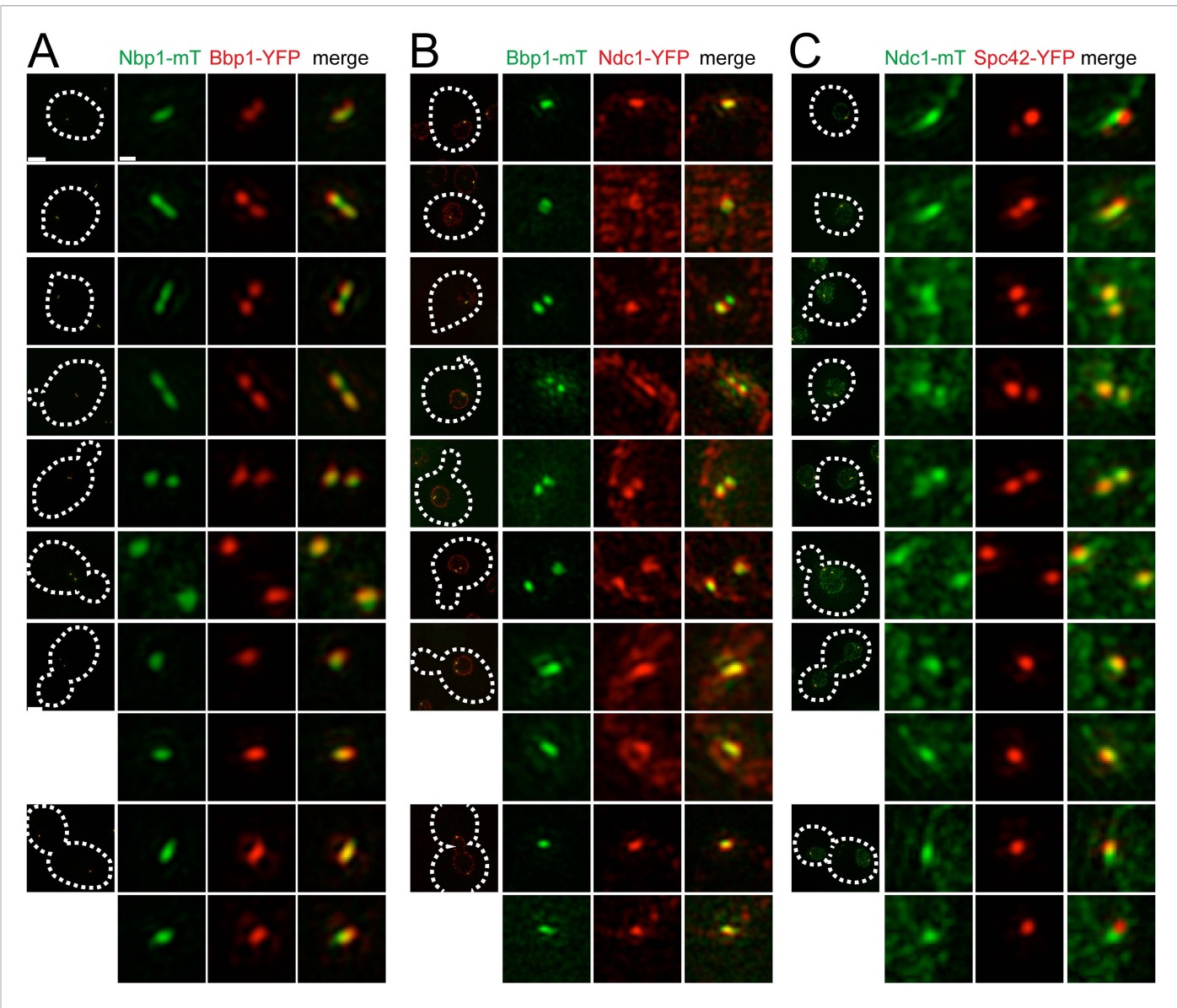

**Figure 7**. Cell cycle analysis of pore membrane component localization. SIM images from asynchronously growing cells containing Nbp1-mTurquoise2 (green) and Bbp1-YFP (red) (SLJ7699, **A**), Bbp1-mTurquoise2 (green) and Ndc1-YFP (red) (SLJ9231, **B**) and Ndc1-mTurquoise2 (green) and Spc42-YFP (red) (SLJ7941, **C**) are arranged based on bud size and distance between SPBs (or satellite structure), which approximates position in the cell cycle. A merged image showing the cell outline (dashes) is shown on the left. Bar, 2 μm. The SPB(s) are shown to the right of each cell. Bar, 200 nm.

## Integrated model of satellite and pore membrane assembly

From our cell cycle analysis of Ndc1-Turquoise2 and Spc42-YFP (*Figure 7C*), we noticed that Ndc1 was slightly displaced from Spc42 along the pole axis, particularly during initiation of SPB assembly (see top and bottom SPBs). As the SPB duplicated, Ndc1-mTurquoise2 appeared to form a layer underneath (towards the nucleoplasm) and later around Spc42, suggesting that the pore membrane forms below the satellite. To better understand the relationship between the satellite and pore membrane components, we applied SPA-SIM to images of α-factor arrested cells containing Spc110-YFP, which localizes to the central plaque of the mother SPB.

As shown in *Figure 8A,B* and *Table 2*, two foci of Mps2, Bbp1, Spc42, Spc29, Nud1, and Cnm67 were observed along the mother-satellite axis, consistent with analysis of individual images (*Figure 5A*, *Figure 5—figure supplement 1A*, *Figure 6C*). Although a bimodal distribution of Ndc1 was also seen, the shallow trough indicates that Ndc1 localized at the mother SPB, satellite and along the length of the extended bridge. Interestingly, Nbp1 was found primarily at the mother SPB

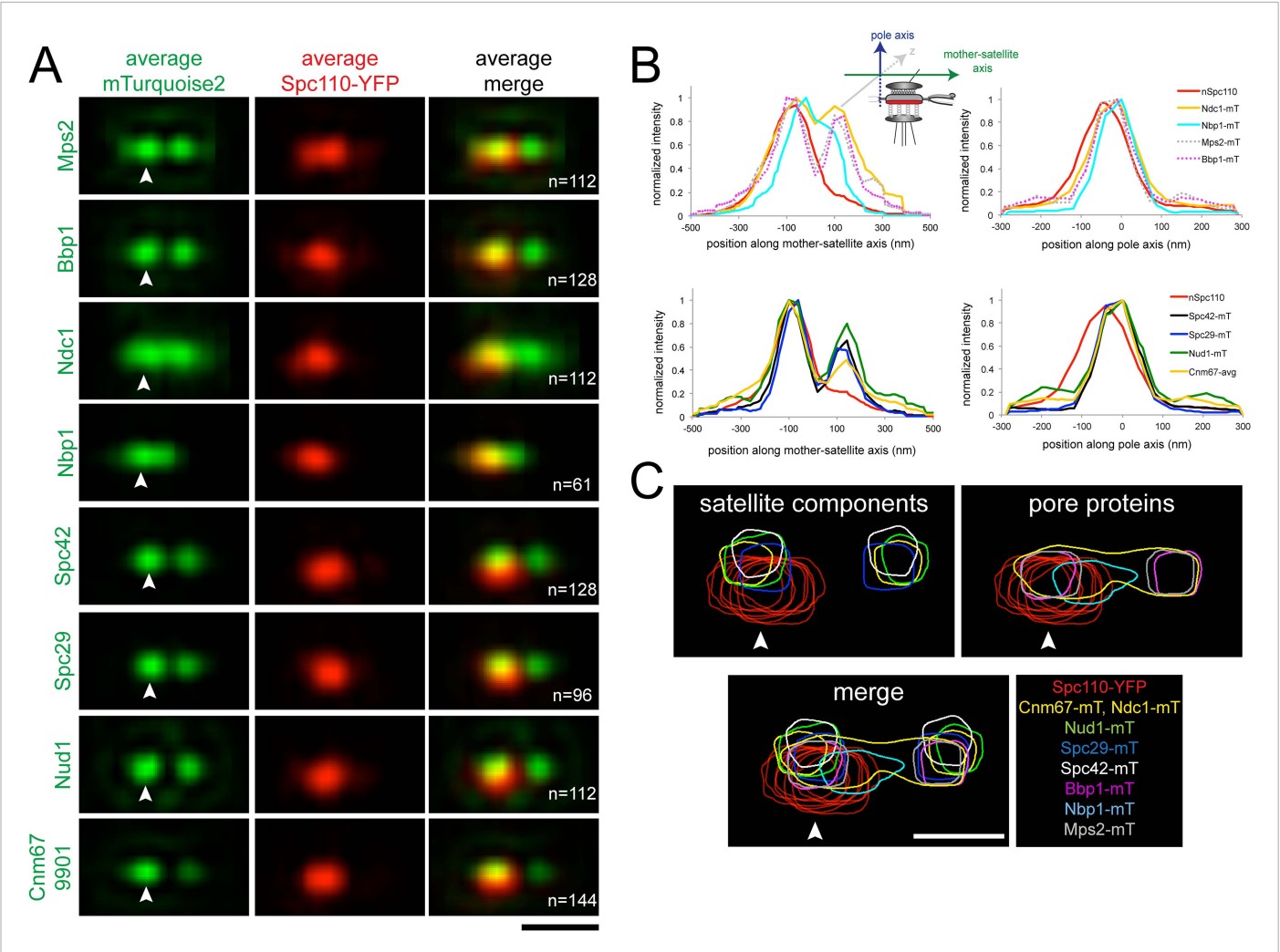

**Figure 8**. Early steps of SPB assembly involve both satellite and pore membrane proteins. (**A**) SIM Images from *Figures 5A, 6C* were aligned and average projections generated as in *Figure 3*. The indicated mTurquoise2 protein (green) or Spc110-YFP (red) at the mother SPB (arrowhead) is shown along with N. Bar, 200 nm. (**B**) Normalized fluorescence intensity of each protein in **A** along the mother-satellite and pole axis, as depicted in the schematic, is plotted using the average Spc110-YFP position from all cells. In these plots, as in *Figure 3*, the center of Spc42-mTurquoise2 was used to define the position of the 0 coordinate in the mother-satellite axis and the pole axis. FWHM values for each are listed in *Table 2*. (**C**) Contour maps showing the distribution of fluorescence intensity at the extended half-bridge based images in **A**. Spc110-YFP in each sample is shown in red and other proteins are colored as indicated. Bar, 200 nm.

**Table 2**. Probability distribution fit parameters from *Figure 8*

| Sample | Query x | y | FWHM$_x$ | FWHM$_y$ | Spc110-reference FWHM$_x$ | FWHM$_y$ |
|---|---|---|---|---|---|---|
| Spc42-mT | −60.3 (0.9) | 0.0 | 73.0 (1.1) | 111.2 (1.5) | 109.4 (1.2) | 144.6 (1.4) |
| | 60.3 (1.2) | | 71.2 (1.8) | | | |
| Spc29-mT | −60.0 (0.9) | −25.8 (1.0) | 70.9 (1.1) | 111.4 (0.9) | 101.5 (1.2) | 157.7 (1.4) |
| | 49.5 (1.3) | | 69.7 (1.8) | | | |
| Nud1-mT | −65.3 (1.2) | −11.6 (1.7) | 78.0 (2.1) | 112.3 (3.8) | 101.8 (1.7) | 148.2 (1.0) |
| | 61.6 (1.8) | | 73.5 (2.6) | | | |
| Cnm67-mT | −60.1 (1.1) | −19.0 (1.6) | 79.9 (1.9) | 91.0 (2.8) | 97.0 (1.3) | 140.1 (1.6) |
| | 74.4 (2.6) | | 61.6 (4.1) | | | |
| Ndc1-mT | −62.7 (2.3) | −34.2 (1.1) | 97.0 (5.6) | 130.9 (1.3) | 106.9 (1.3) | 137.4 (1.4) |
| | 56.0 (3.3) | | 56.2 (5.4) | | | |
| Nbp1-mT | −66.5 (1.2) | −48.8 (1.0) | 77.7 (4.3) | 106.8 (1.1) | 102.1 (1.0) | 119.1 (1.0) |
| | 3.0 (1.8) | | 50.6 (2.3) | | | |
| Mps2-mT | −63.1 (1.9) | −37.2 (1.4) | 60.0 (6.4) | 96.4 (2.1) | 146.6 (3.3) | 132.8 (2.0) |
| | 42.8 (3.0) | | 47.2 (6.2) | | | |
| Bbp1-mT | −53.8 (1.5) | −35.8 (1.5) | 87.5 (3.2) | 108.2 (2.2) | 113.0 (2.3) | 137.6 (2.6) |
| | 62.9 (2.2) | | 65.3 (2.8) | | | |

Notes: All values are in nm. In all cases 'x' refers to the mother-satellite axis and 'y' refers to the pole axis. The origin is at the center point between the C-terminal Spc42 peaks as in *Figure 3*. The Spc110 distributions are all centered at −44.2 (1.3) in x and −61.1 (0.8) in y. For Nbp1, Ndc1, and Mps2 a third Gaussian centered on the half-bridge with a broad width was added to allow for proper Gaussian fitting of the visible peaks. As in *Table 1*, the FWHM is 2.35 times the standard deviation of the Gaussian fit, and this can be converted into 95% integral values by multiplying each by 1.7. In all cases the mother SPB peak is shown on the first line and the satellite peak, if applicable, is shown on the second line. FWHM$_y$ in cells containing two foci was determined by the averages over the two peaks. Errors are in parentheses and are standard deviations from Monte Carlo random fits as described in 'Materials and methods'.

and in bridge region proximal to the mother SPB in α-factor arrested cells. The fact that Nbp1 is the only pore membrane component not present at the satellite suggests that it may be a limiting factor restricting membrane insertion, or that its recruitment to the new SPB may require a maturation event that has not yet occurred.

Analysis of positional information obtained from the Gaussian fits used for SPA-SIM (*Table 2*) showed that Nbp1-mTurquoise2 is located closest to the Spc110-YFP plane, whereas Ndc1-mTurquoise2, Mps2-mTurquoise2, and Bbp1-mTurquoise2 are 24–27 nm away. The peaks of Spc29-mTurquoise2, Spc42-mTurquoise2, Cnm67-mTurquoise2, and Nud1-mTurquoise2 are further displaced towards the cytoplasm in the pole axis by 35, 61, 42, and 50 nm, respectively. The location of these satellite components, with the exception of Spc42, relative to the C-terminus of Spc110 is consistent with our knowledge of their positional information within the core SPB from other methods of analysis such as immunoEM, FRET, and yeast two-hybrid analysis (*Adams and Kilmartin, 1999*; *Jaspersen and Winey, 2004*; *Muller et al., 2005*; *Winey and Bloom, 2012*). The large displacement of Spc42-mTurquoise2 is most likely due to the fact that the fits consider the position of Spc42 at both the mother SPB and satellite. The shift of the satellite components relative to the membrane proteins indicates that the pore and pore-related structures are formed near the region defined by the C-terminus of Spc110, which is displaced in the pole axis from the satellite components. Overlays of fluorescent density further illustrate how pore proteins assemble beneath the satellite (*Figure 8C*).

## Nbp1 and Ndc1 localization during membrane insertion

Release of cells from α-factor into the cell cycle allowed us to compare the kinetics of Spc110 acquisition (which occurs at the end of duplication after the new SPB is inserted into the NE) with that

of incorporation of Nbp1 or Ndc1 to determine if Nbp1 is indeed a limiting factor for SPB duplication as our SPA-SIM analysis suggested. Continued growth of cells allowed us to watch NE insertion and protein incorporation during the next cell cycle to ensure the α-factor arrest did not affect recruitment of Nbp1 or Ndc1.

A single Spc110-YFP focus and a single Ndc1-mTurquoise2 or Nbp1-mTurquoise2 focus was observed in Ndc1-mTurquoise2/Spc110-YFP or Nbp1-mTurquoise2/Spc110-YFP strains arrested in α-factor (*Figure 6C*) and at the 5 min time point following release (*Figure 9A,B*). At 15 or 20 min following release, two foci of Ndc1-mTurquoise2 were detected in 32% (44/136) or 34% (37/110) of cells, respectively. In these cells, a single Spc110-YFP focus was observed, providing evidence that Ndc1 localizes to the new SPB before insertion into the NE (*Figure 9B,D*). In contrast, only 9% (12/129) or 10% (17/165) of cells at 15 or 20 min had two foci of Nbp1-mTurquosie2 with a single focus of Spc110-YFP. Most (88%, 114/129 at 15 min and 79%, 104/165 at 20 min) contained a single focus of Nbp1-mTurquoise2 and a single focus of Spc110-YFP. Even at later time points, the predominant phenotype seen was two resolved foci of Spc110-YFP that likely are SPBs in the duplicated side-by-side configuration connected by an elongated Nbp1-mTurquoise2 signal (18%, 19/107 at 25 min and 38%, 24/65 at 30 min) (*Figure 9B,D*). It is important to note that we used mTurquoise2, which has a shorter wavelength than YFP (and thus increased resolution with SIM) to visualize Nbp1. The fact that we saw two foci of Spc110-YFP suggests that biology inherent to SPB duplication, rather than the resolution of our microscope, was the underlying cause for this stretched configuration of Nbp1-mTurquoise2. A similar pattern of Ndc1-mTurquoise2 recruitment to the new SPB prior to its insertion and Nbp1-mTurquoise2 resolution after SPB insertion was observed during the next SPB duplication cycle (*Figure 9A,B*), although precise timing of events relative to release from α-factor varied from cell to cell due to loss of synchrony following mitosis. Although two resolved foci of Nbp1-mTurquoise2 occasionally were seen prior to NE insertion, these events were rare (less than 10% of cells examined at all time points) and could be due to poor detection of Spc110-YFP. Taken together with our SPA-SIM results, these data lend support to the idea that Nbp1 is the last SPB component to fully assembly onto the new pole. We propose that a membrane fenestra formed by Mps2 and Bbp1 is created below the satellite at the time of its formation and that Ndc1 is recruited later followed by Nbp1 (*Figure 10*).

## Discussion

SPA-SIM allowed us to resolve features of the extended half-bridge formed by Sfi1 and other half-bridge components that were not observed using EM, biochemical, genetic, or other super-resolution methods. A recent study that combined PALM and STORM data showed, as we have, that N-terminally tagged versions of Sfi1 appear as two foci and a single focus is observed if Sfi1 is tagged at its C-terminus (*Seybold et al., 2015*). However, using SPA-SIM we were able to resolve additional structural features of the extended half-bridge due to our ability to perform quantitative analysis of multiple SPB components in three dimensions. Unexpectedly, we find that the bridge is neither linear nor symmetric, even at early stages of SPB duplication, and that Cdc31 is not localized uniformly along its length. The maximum intensity of Cdc31 binding on the extended half-bridge occurs in the central region of the new Sfi1 filament, which is located between the central and distal bridge (*Figure 3E*). While this central repeat domain of Sfi1 has been shown in vitro to contain multiple Cdc31 binding sequences (*Kilmartin, 2003*; *Li et al., 2006*; *Seybold et al., 2015*), it is unknown why Cdc31 would preferentially associate with the newly formed Sfi1 instead of the old molecules of Sfi1 since both contain the same repeat domains. By EM, the half-bridge appears as a flat sheet and probably only has one layer of Sfi1 filaments (*Byers, 1981*; *Li et al., 2006*). Based on the 65° rotation of Cdc31 molecules around Sfi1 repeats seen in vitro, there is thought to be limited side-to-side contact between adjacent Sfi1-Cdc31 filaments (*Li et al., 2006*). Cdc31 binding to only a fraction of repeats, in particular those on the new filament, could increase the interaction between adjacent Sfi1 molecules by partially alleviating this constraint. Binding to the newly assembled Sfi1 also could be important for stabilizing the half-bridge, a theory supported by recent studies on *sfi1+* and *cdc31+* in fission yeast (*Lee et al., 2014*; *Bouhlel et al., 2015*).

Initiation of a new round of SPB duplication requires elongation of the half-bridge. During most of the cell cycle, the C-terminus of Sfi1 is phosphorylated; dephosphorylation of Cdk1 sites in the C-terminus by Cdc14 results in half-bridge elongation and triggers SPB reduplication (*Avena et al., 2014*). Here we show that two foci of GFP-Sfi1 are visible in late mitosis, when Cdc14 is activated (*Stegmeier and Amon, 2004*). Based on our analysis, this appears to be the first landmark event signaling a new round of SPB duplication. Understanding the molecular interactions between the C-termini of Sfi1 molecules

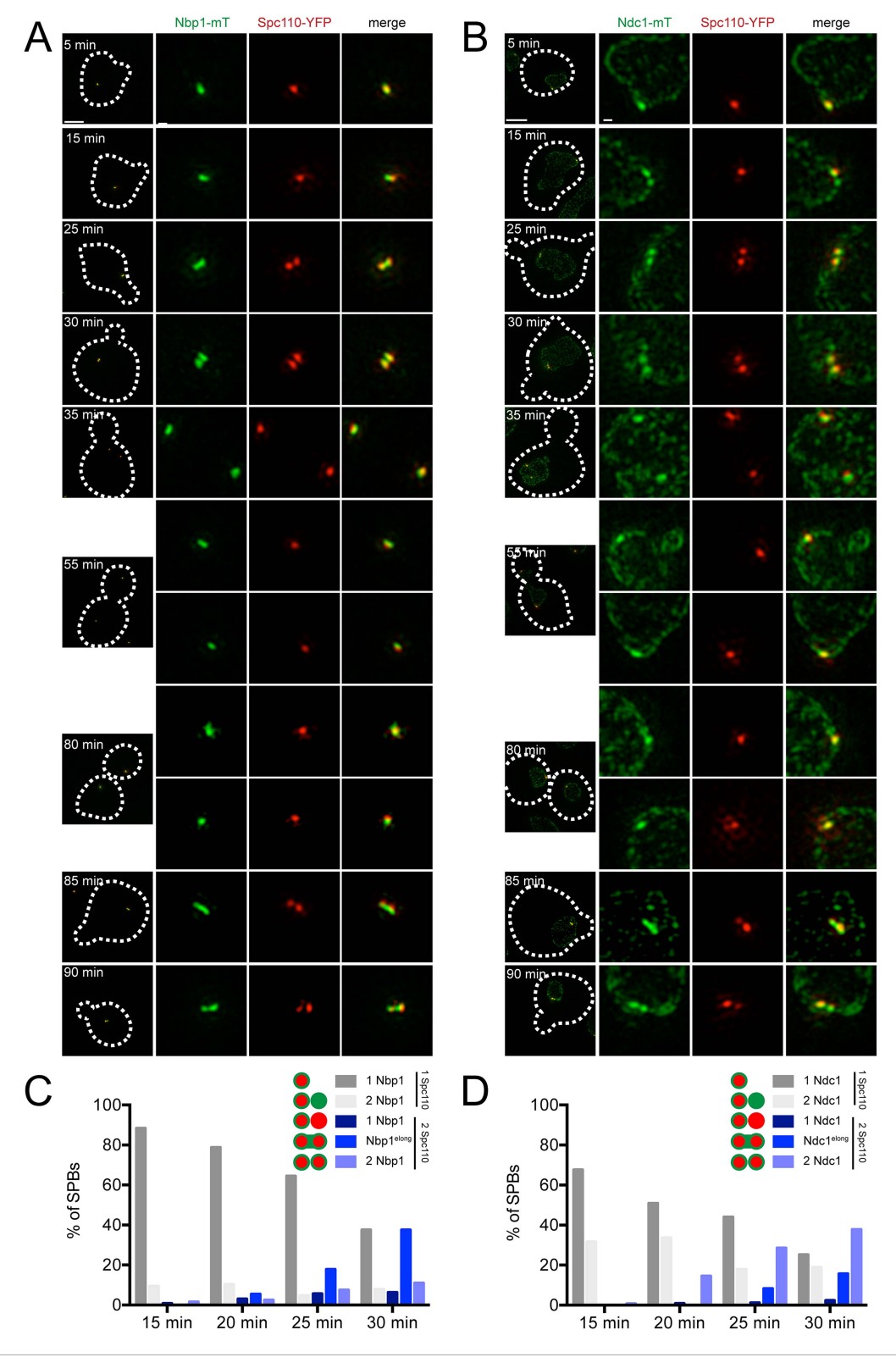

**Figure 9**. Localization of Ndc1 and Nbp1 during SPB insertion. Cells containing Nbp1-mTurquoise2 (**A** and **C**, SLJ10169) or Ndc1-mTurquoise2 (**B** and **D**, SLJ10018) along with Spc110-YFP were released from α-factor into prewarmed SC-complete media at 30°C. At 5 min time intervals, an aliquot of cells was harvested, fixed and imaged by SIM. (**A**, **B**) Merged images showing the outline (dashes) of a representative cell is on the left. Bar, 2 μm.
*Figure 9. continued on next page*

*Figure 9. Continued*

The SPB(s) are to the right of each cell. Bar, 200 nm. (**C**, **D**) Cells at the 15, 20, 25, and 30 min time points were analyzed to determine the percentage of schmooed and small budded cells that have two closely spaced foci of Spc110-YFP and/or Nbp1-mTurquoise2 (**C**) or Ndc1-mTurquoise2 (**D**). Schematics depict the five configurations of proteins, including the signal that was stretched between mother and daughter SPB.

emanating from the same pole and between Sfi1 C-termini in the antiparallel array will further elucidate how this important step in SPB duplication is controlled.

Our finding that Bbp1 and Mps2 localize to the satellite region early in SPB duplication was unexpected; however it could explain, in part, how satellite components are anchored to the bridge and NE. Not only do Mps2 and Bbp1 interact with each other, but also Bbp1 binds to Spc29 and Kar1 and Mps2 associates with Mps3 (*Schramm et al., 2000*; *Jaspersen et al., 2006*). Enrichment of Mps3-YFP at the distal end of the bridge near the satellite (*Figure 3E,F*) could be due to interactions with Mps2. The *mps2-381* allele that is defective in Mps3 binding is unable to form a bridge or satellite at the non-permissive temperature, supporting the idea that Mps2 plays a role early in SPB duplication (*Jaspersen et al., 2006*). A role for Bbp1 early in SPB duplication has not been reported; however, the observation that Mps1-dependent phosphorylation of Spc29 increases its binding to Bbp1 (*Araki et al., 2010*) might be re-interpreted in light of our finding—phosphorylated Spc29 may recruit Bbp1 to the satellite. Similarly, other regulatory events at the SPB may take on a new significance based on possible coordination between satellite and pore assembly.

If pore proteins localize to the satellite region in α-factor arrested cells, why does the SPB not insert into the NE? The answer likely is that factors needed to remodel the NE to form a stable contiguous pore membrane have not yet localized. One candidate is Ndc1, an evolutionarily conserved integral membrane protein that is present at the pore membrane of both SPBs and NPCs (*Chial et al., 1998*). Although Ndc1 was observed at the satellite region in α-factor arrested cells, yeasts are exquisitely sensitive to its dosage (*Chial et al., 1999*; *Chen et al., 2014*) and it may not be present at high enough levels to induce stable changes in membrane architecture. A second candidate is the Ndc1 binding protein Nbp1, which contains an ArfGAP1 lipid packing sensor domain that is able to interact with lipid head groups to induce membrane bending (*Bigay et al., 2005*; *Kupke et al., 2011*). Nbp1 localization in α-factor arrested cells was restricted to the mother SPB and the bridge proximal to the mother. Our observations that Nbp1 does not localize to the new SPB until that pole is inserted into the NE and that it localizes to a region within 10 nm of Spc110 are consistent with the idea that Nbp1 functions from the nucleoplasm (*Kupke et al., 2011*). Analysis of cells depleted of *NBP1* using a temperature-dependent degron allele showed that Nbp1 is not required for localization of Cdc31, Kar1, Mps3, Sfi1, Spc29, or Spc42 to the mother SPB or the satellite. However, Nbp1 is partially required for Bbp1 recruitment and Mps2 stability at the new SPB without significantly affecting the

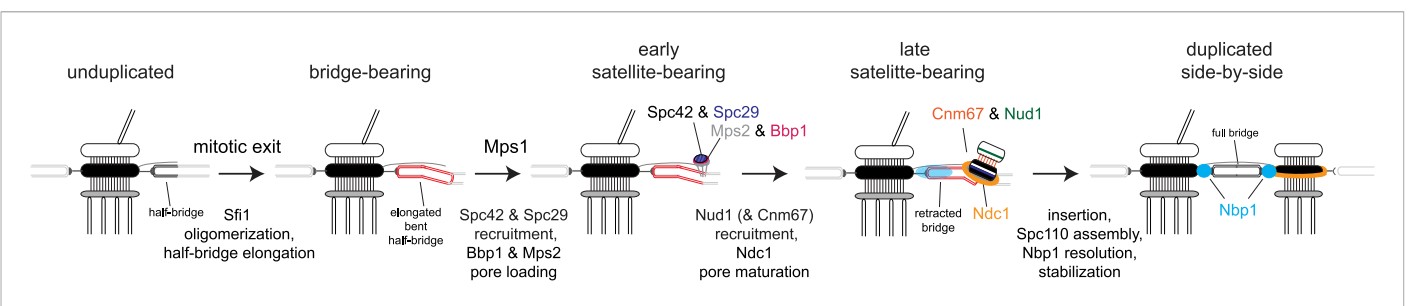

**Figure 10**. Model for SPB duplication. Revised model of the early steps in SPB duplication based on results described. In anaphase, Sfi1 oligomerization leads to an elongated half-bridge. Next, Spc42 and Spc29 assemble at the distal cytoplasmic tip of the bridge. At the same time, the pore proteins Bbp1 (magenta) and Mps2 (gray) accumulate at the satellite region. Later, the satellite and membrane-associated region continue to mature through the addition of Nud1 (green), Cnm67 (orange), and Ndc1 (orange ring). Nbp1 (cyan) remains associated primarily with the mother-proximal end of the extended half-bridge until the SPB is inserted into the NE. Note that satellite and pore-associated proteins are also present at the mother SPB.

mother SPB (*Araki et al., 2006*). The fact that SPB insertion can occur without full resolution of Nbp1 onto a new SPB combined with its requirement for Mps2 protein stability make it tempting to speculate that Nbp1 is not required for SPB insertion per se but rather for maturation and maintenance of structures formed at the membrane of the new SPB.

Cnm67 links Nud1 and Spc72 at the outer plaque with Spc42 at the central plaque (*Adams and Kilmartin, 1999*; *Schaerer et al., 2001*; *Muller et al., 2005*). Cells lacking *CNM67* display defects in cytoplasmic microtubule organization but remain viable, undergoing a normal SPB duplication cycle (*Brachat et al., 1998*; *Hoepfner et al., 2000*; *Schaerer et al., 2001*). Our observation that Cnm67 is present at variable levels at the satellite is consistent with the idea that it is not required for SPB duplication. In cells lacking *CNM67*, ~10% of cells showed Nud1 localization to the half-bridge region, suggesting that there is a Cnm67-independent mechanism of SPB localization (*Adams and Kilmartin, 1999*). This could explain, in part, why Nud1 is detected at the satellite region more frequently than Cnm67 in our experiments. Although Nud1 is an essential component of the outer plaque and is present at the satellite, it has no known role in SPB duplication. Instead, Nud1 serves as a signaling platform that is important for exit from mitosis (*Adams and Kilmartin, 1999*; *Gruneberg et al., 2000*; *Rock et al., 2013*). However, given that detection of Cnm67 at the satellite was challenging and varied considerably between strains, our results may underestimate levels of Cnm67 at the satellite.

Our ability to determine the spatiotemporal relationship of multiple SPB components was integral to the new insights in SPB structure and duplication described here. While SIM by itself does not offer the resolution of STORM and PALM, the SPA-SIM method we developed offered unique advantages that were ideally suited to our study. Firstly, as opposed to STORM and PALM, SIM data can be acquired using standard fluorescent proteins such as GFP, YFP, CFP, mCherry, and mTurqouise2. In yeast, where genetic manipulations are straightforward, this allows for study of endogenously tagged proteins and removes concerns over competition with native untagged proteins as well as the need for antibody staining. Because all protein can in theory be visualized, sparse labeling artifacts are effectively eliminated, which pose a major challenge for any type of quantitative single particle analysis. Secondly, SIM allows for simple and rapid measurement with two to three fluorophores. In our case, we have used this to our advantage since Spc42 or Spc110 could be included as reference points for further alignment and averaging. Our fitting uncertainty of Gaussian centers in three dimensions is less than 3 nm, while the alignment of the colors, shown for YFP-Spc42-mTurqouise2 (*Figure 3*), is within $12 \pm 6$ nm in the mother-satellite axis, which serves as a color alignment control for our SIM data and our realignment method. Furthermore, we are able to decipher shifts in density centers on the order of 10–30 nm, putting SPA-SIM on par with other super-resolution and particle averaging methods that to this point have required more sophisticated three-dimensional data acquisition methods (*Loschberger et al., 2012*; *Szymborska et al., 2013*; *Loschberger et al., 2014*; *Van Engelenburg et al., 2014*; *Broeken et al., 2015*).

In summary, we have used SIM and SPA-SIM to uncover important mechanistic details about a process that has been extensively studied genetically, cytologically, and biochemically. Our localization analysis of wild-type proteins provides a dynamic picture of the process of SPB duplication that can be used in the future for the study of cell cycle and SPB mutants. Furthermore, our experimental methods can be applied to multiple biological structures and thus provide a framework for how advanced microscopy methods can be applied to elucidate details of biological systems that cannot be captured in vitro.

## Materials and methods

### Yeast strains

Yeast strains are derivatives of W303 and are listed in *Source data 1*. Fusions to GFP, YFP, CFP, mCherry, and mTurquoise2 were created using polymerase chain reaction-based methods in SLJ1070 (*Mata/Matα bar1/bar1 ade2-1/ADE2 trp1-1/TRP1 lys2Δ/LYS2 leu2-3,112/leu2-3,112 his3-11,15/his3-11,15 ura3-1/ura3-1*) (*Gardner and Jaspersen, 2014*). Haploid strains were generated by sporulation and tetrad dissection. *MET3-CDC20-KANMX-HO* (a gift of Marc Gartenberg, Rutgers) was integrated into strains by digestion with AflII and transformants were selected on plates containing G418. The N- and C-terminal-tagged versions of *SFI1* were gifts of John Kilmartin (University of Cambridge) and were integrated into strains as described (*Avena et al., 2014*). *mps1-1* (*Straight et al., 2000*) was crossed into

these strains. *MET3-YFP-CDC31* and several YFP- and CFP-tagged versions of *CNM67*, *SPC42*, *SPC29*, and *SPC110* were gifts from Trisha Davis, Eric Muller, and Tennessee Yoder (University of Washington).

Strains were grown in minimal media supplemented with 3× adenine to mid-log phase at 30°C, with the exception of strains containing *mps1-1*, which were grown at 24°C then shifted to 37°C for 4 hr. To arrest cells in G1, 1 µg/ml or 10 µg/ml α-factor was added to *bar1* or *BAR1* strains, respectively, for 3 hr at 30°C. To arrest cells in metaphase, strains containing *MET-CDC20* were grown in YPD for 2.5–3 hr at 30°C. Following three washes, cells were released into minimal media lacking methionine. 1 µg/ml α-factor was added to half of the released culture; dimethyl sulfoxide was added to the other half. Both were incubated at 30°C and samples were taken at the indicated times.

## SIM imaging

Cells were fixed for 15 min in 4% paraformaldehyde (Ted Pella) in 100 mM sucrose and then washed two times in phosphate-buffered saline, pH 7.4. An aliquot of cells was placed on a glass slide and covered with a number 1.5 coverslip. SIM images were acquired with an Applied Precision OMX Blaze (GE Healthcare). A 60× 1.42 NA Plan Apo oil objective was used, and emission was collected onto two PCO Edge sCMOS cameras (Kelheim, Germany) with each camera dedicated to one specific channel. For CFP/mTurqouise2/YFP experiments, a 440/514/561 dichroic was used with 460–485 nm and 530–552 nm emission filters for CFP/mTurqouise2 and YFP, respectively. The 440 nm line and 514 nm line were used for excitation of CFP/mTurqouise2 and YFP, respectively. For GFP/mCherry experiments, a 405/488/561/640 dichroic was used with 504–552 nm and 590–628 nm emission filters for GFP and mCherry, respectively, using a 488 nm laser line (GFP) and 561 nm laser line (mCherry). To limit spectral cross-talk, all SIM data was acquired in alternating excitation mode. SIM reconstruction was performed with the Applied Precision software Softworx with a Wiener filter of 0.001. Color alignment from different cameras in the radial plane was performed using the color alignment slide from GE Healthcare (Pittsburg, PA). In the axial direction, color alignment was performed using 100 nm tetraspeck beads (Life Technologies, Kelheim, Germany). For image preparation, the SIM reconstructed images were scaled 2 × 2 with bilinear interpolation then smoothed with a Gaussian blur of pixel radius 0.8. In many cases, for illustration purposes, a max projection in z over the relevant slices was done.

## SPA-SIM

Three-dimensional analysis of SIM images was performed with custom written macros and plugins in the open source software, ImageJ (NIH, Bethesda, MD). All plugins and their source code are available at http://research.stowers.org/imagejplugins/. In cases where the protein of interest represents two distinct spots corresponding to the mother SPB and satellite, the spots were fitted to two 3D Gaussian functions (see below) and realigned along the axis between these functions for further analysis and visualization. In cases where the protein represents a less distinct distribution or when the distribution along the mother-satellite axis is being queried, a secondary protein (typically Spc42) was fit to two 3D Gaussian functions for further analysis and visualization.

SPBs were identified in a semi-automated fashion by sum projecting the raw SIM images, blurring them with Gaussian blur with a standard deviation of 1 pixel and progressively finding all maxima with an intensity above 15% of the image maximum intensity and a minimum distance of 30 pixels from other maxima. Next 30 × 30 pixel intensity profiles along the z axis were created. SPBs with fitted channel profile maxima in the first or last slice were eliminated. Each selected SPB was then visually inspected in three dimensions for counts of those containing two spots. This methodology underestimates the number of two spot images given the lower z resolution and thus the inability of the microscope to distinguish spots where the mother-satellite axis is oriented vertically. Nevertheless, yeast nuclei rarely orient themselves this way and this method provides the best estimate of satellite formation and/or incorporation.

SPBs containing two visible spots were selected for fitting initialization with the mother selected first. In samples with Spc110 labeling, the spot closest to the Spc110 was identified as the mother SPB. In samples where Spc42 was used for fitting, the brighter Spc42 spot was considered the mother SPB, as was shown in Spc42/Spc110 dual labeling experiments (see *Figure 5A,B*). While this method has

some associated error, most proteins are distributed fairly symmetrically along the mother-satellite axis so a small number of mis-oriented axes will not dramatically skew the results.

After manual spot selection, the spots were fit to the sum of two 3D Gaussian functions by Levenberg–Marquardt non-linear least squares (*Bevington and Robinson, 2003*). Spot centers in x and y were constrained to within ± two pixels of the manually selected positions to avoid dramatic misfits. The center position in z was initialized as the maximum slice of a 3 × 3 pixel z profile and also constrained to ± two slices.

After fitting, a two dimensional plane profile in both channels was created containing the fitted centers of the spots and oriented horizontally along the axis perpendicular to the mother-satellite axis. Intensities were obtained by tri-linear interpolation along this plane with a pixel size four times smaller than that of the reconstructed SIM image. These realigned profiles were summed together to create dual-color probability maps for the intensity profiles of the two proteins under study. In cases where Spc110 is labeled in the second channel, profiles were rotated so that the side of the profile with the higher Spc110 intensity (the mother SPB) was oriented to one side. In that way it was possible to assess the shift of the mother-satellite profile relative to the pole axis. In the same way, half-bridge profiles collected with Spc42 as a reference were oriented to one side based on the non-Spc42 signal (or YFP in the YFP-Spc42-mTurquoise2 strain). Typically images are shown with the aligned channel above the non-aligned channel since this is most consistent with available structural and molecular information. The exceptions to this are Mps3 and N-terminus of Spc42, where EM and/or FRET data demonstrates positioning below the mother-satellite axis (*Bullitt et al., 1997*; *O'Toole et al., 1999*; *Jaspersen et al., 2002*; *Muller et al., 2005*).

Analysis of probability map centers for distance and angle measurements were made by multi-Gaussian fitting of average intensity profiles along either axis. Errors were estimated by standard Monte Carlo analysis of 100 randomly simulated data sets with random Gaussian noise corresponding to the variance indicated by the fit residuals (*Bevington and Robinson, 2003*). Given the random orientation of individual realigned images and the choice to orient them with the reference channel to one side, the mother-spindle axis was centered slightly differently for different probability maps. For creation of overlayed contour maps, these differences were eliminated by vertically shifting the probability maps with bilinear interpolation. Contour maps were then generated by thresholding each channel at 75% of its maximum intensity and outlining the resulting mask. For proteins present at both the mother and satellite, each distribution was outlined independently to avoid bias due to the intensity differences between these distributions.

Centers and FWHM values for the probability distributions were generated by fitting average horizontal and vertical profiles to one-dimensional Gaussian functions. For horizontal (x direction) half-bridge profiles with a single peak, 30 pixels at either end of the distribution were removed to prevent mis-fitting due to variable background signals in that region. For pore protein distributions in that same direction, a third broad Gaussian function centered on the distribution was required for reasonable fitting of the distribution.

## EM

Log-phase cells were high-pressure frozen in a Wohlwend Compact 02 high pressure freezer, freeze-substituted in 0.25% glutaraldehyde, 0.1% uranyl acetate in acetone, embedded in Lowicryl HM20, and processed for immunoEM as previously described (*Giddings et al., 2001*). The affinity-purified rabbit polyclonal GFP antibody was a gift from Jason Kahana and Pam Silver. Imaging was conducted using a FEI Phillips CM100 electron microscope.

## Acknowledgements

We thank Trisha Davis, Tennessee Yoder, John Kilmartin, Marc Gartenberg and Shelly Jones for plasmids and strains and Jason Kahana and Pam Silver for a GFP antibody. We thank Alex Stemm-Wolf for EM strain assistance, Tom Giddings Jr for EM sample preparation and imaging and Ann Cavanaugh and Jennifer Gardner for help with strains and SIM imaging. We are grateful to the P01 group and the Jaspersen and Winey labs for helpful suggestions and comments on the manuscript. JA and MW are funded by NIH R01 GM51312 (to MW), P01 GM105537 (M Winey, PI) and NIH 5 T32 GM007135 (to JA) and the Jaspersen lab is supported by the Stowers Institute for Medical Research.

## Additional information

### Funding

| Funder | Grant reference | Author |
|---|---|---|
| National Institutes of Health (NIH) | P01 GM105537 | Mark Winey |
| National Institutes of Health (NIH) | 5 T32 GM007135 | Jennifer S Avena |
| National Institutes of Health (NIH) | R01 GM51312 | Jennifer S Avena, Mark Winey |
| Stowers Institute for Medical Research (SIMR) | | Shannon Burns, Sue L Jaspersen |

The funders had no role in study design, data collection and interpretation, or the decision to submit the work for publication.

### Author contributions

SB, Read and approved article, Conception and design, Acquisition of data, Analysis and interpretation of data; JSA, BDS, Conception and design, Acquisition of data, Analysis and interpretation of data, Drafting or revising the article; JRU, SLJ, Conception and design, Analysis and interpretation of data, Drafting or revising the article; ZY, Read and approved article, Essential for acquisition of all data, Acquisition of data; SES, Read and approved article, Analysis and interpretation of data; MW, Conception and design, Drafting or revising the article

## Additional files

### Supplementary file

• Source data 1. Yeast strains.

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
