## [Decision Letter]

Thank you for submitting your work entitled “Structured-illumination with particle averaging reveals novel roles for yeast centrosome components during duplication” for peer review at *eLife*. Your submission has been favorably evaluated by Randy Schekman (Senior Editor), Karsten Weis (Reviewing Editor), and two reviewers.

The reviewers have discussed the reviews with one another, and the Reviewing Editor has drafted this decision to help you prepare a revised submission.

In this manuscript, Burns et al. use SIM and single particle averaging (SPA-SIM) in order to characterize molecular events associated with SPB duplication. Overall, the reviewers felt that this is an excellent manuscript employing state-of-the-art imaging to gain insight into the structure and assembly of spindle pole bodies. The article is well-written and, in general, the data support the authors’ conclusions. However, there were some concerns that the results remain somewhat descriptive and it was concluded that additional experiments would be needed.

1) The authors propose that Nbp1 is a limiting component for SPB insertion. The study would be tremendously strengthened if they could actually show this using mutants or degron-fusion proteins for pore components and analyzing the recruitment and appearance of other pore and satellite proteins.

2) The authors use *CDC20* to arrest and then release cells. While this is a commonly employed technique for biochemical analyses in the field, it is not clear why this is important for single cell analysis. The authors could use a morphological marker (such as pole separation) to tell them exactly where they are within the cell cycle. The issue here is how representative arrest and release is relative to “normal” cell cycle progression. One might imagine a build-up of certain products in the arrest that might be different from normal progression. It is not necessary that the authors repeat their entire data set but they should validate for a few key proteins that the changes they see are representative of unperturbed cell cycles.

3) The description of the methods is often not sufficient to understand the details. For example, while it is clear how the y-axis is determined, it remains obscure how the authors establish the x-axis. Here, more explanation is required.

4) For the position of proteins from the single particle averaging, the distribution of maximum intensity is provided, with the standard error of the mean indicated. The authors need to include a table indicating additional structural information. This should include the FWHM measurement for each protein. Note in Figure 3 that the distribution of Sfi1 is greater (in width) than Kar1. It is important to have a table with these statistics. In addition, the area over which these proteins occupy could be expressed as the area in which 95% of fluorescence is contained. This would help the reader appreciate the distribution, perhaps with more insight than simply the maximum intensity.

5) The manuscript lacks precision in several places and the authors should carefully edit their Results section. For example, in Figure 5, around 70% of pheromone arrested cells show two Nud1 foci (SPB and satellite), whereas in Figure 5, this number drops to less than 10%. Why? Is this due to a different duration of pheromone arrest? The authors should give this duration for 5A and B either in the main text or in the figure legend.

Also, in the main text, the description of the data in Figure 5 does not mention that the cells were arrested with pheromone, which makes the reading confusing. Furthermore in 5A and B, while Cnm67 is thought to link Nud1 to the SPB, it is absent from the satellite in more than 60% of the cells where Nud1 is present around 70% of the time. What do the authors make of that? There are many examples of similar issues.

In Figure 4, the vast majority of non-arrested cells show a single SPC42 focus associated with a single Sfi1 focus. Does that mean that ∼90% of these cells are in metaphase? In the *mps1-1* arrested cells, cells show many more configurations. Does it mean that the cells arrest at different points in the duplication cycle?

In Figures 2 and 3, the authors suggest the existence of a 17° kink in the bridge. Along which axis is this kink? Could this be simply due to the fact that the bridge is on the membrane while Spc42 is inside the pore?

---

## [Author Response]

*In this manuscript, Burns et al. use SIM and single particle averaging (SPA-SIM) in order to characterize molecular events associated with SPB duplication. Overall, the reviewers felt that this is an excellent manuscript employing state-of-the-art imaging to gain insight into the structure and assembly of spindle pole bodies. The article is well-written and, in general, the data support the authors’ conclusions. However, there were some concerns that the results remain somewhat descriptive and it was concluded that additional experiments would be needed*.

We thank the reviewers and editors for their consideration of our manuscript and their positive comments. We have revised the paper to ensure that our methods and results are clearly and accurately presented to justify our conclusions. We recognize that the work may seem descriptive, but analysis of 18 proteins in wild-type cells resulted in findings that we believe significantly advance our understanding of SPB duplication.

*1) The authors propose that Nbp1 is a limiting component for SPB insertion. The study would be tremendously strengthened if they could actually show this using mutants or degron-fusion proteins for pore components and analyzing the recruitment and appearance of other pore and satellite proteins*.

We agree with the reviewers about the value of mutants, and we have included analysis of *mps1-1* in Figure 4. We also have done multiple timing experiments to determine how SPB components are recruited during the course of wild-type SPB duplication to build a complete model of the duplication process using the lens of SIM. To show that Nbp1 is indeed the last SPB component to fully assemble on the new SPB, we have added new data in which we released cells from α-factor and examined its distribution relative to Spc110, which marks assembly into the NE. Unexpectedly, we observed two foci of Spc110 before we detected two Nbp1 foci (Figure 9). This supports our idea that Nbp1 is the last and/or limiting factor but also demonstrates that resolution of Nbp1 into two foci is not required for SPB insertion.

Previously, we have examined protein localization in the Nbp1 degron strain, finding that in addition to Nbp1, Bbp1 and Mps2 are lost at a fraction of SPBs; Cdc31, Kar1, Mps3, Sfi1, Spc42 and Spc29 are unaffected (Araki et al., MBoC, 2006). SIM is unlikely to tell us more than this. We also showed in this same paper that Nbp1 localization is affected in some *mps2* mutant strains.

Unfortunately, degrons and many mutant alleles in SPB components destroy SPB structure, and it is challenging or impossible to sort out primary versus secondary effects based solely on localization data. Precision tools that disrupt individual interactions between pore proteins and each other or satellite components do not exist. Mutants with specific defects defining a single function of Nbp1 would require extensive work to produce and characterize, warranting a separate report beyond the scope of this paper. We feel our solid, systematic characterization of the 18 components in wild-type cells significantly raises the knowledge base in the field and will be a springboard for many future studies.

*2) The authors use* CDC20 *to arrest and then release cells. While this is a commonly employed technique for biochemical analyses in the field, it is not clear why this is important for single cell analysis. The authors could use a morphological marker (such as pole separation) to tell them exactly where they are within the cell cycle. The issue here is how representative arrest and release is relative to “normal” cell cycle progression. One might imagine a build-up of certain products in the arrest that might be different from normal progression. It is not necessary that the authors repeat their entire data set but they should validate for a few key proteins that the changes they see are representative of unperturbed cell cycles*.

While we agree that the distance between spindle pole bodies is a useful method to determine the position the cell cycle, our study is the first to examine early SPB duplication structures using fluorescence microscopy, so what to use as a reference point to stage this period of pole duplication was unknown. There is certainly a precedent that SPB components may build up at the pole during a *cdc20* arrest (O’Toole et al., MBoC, 1997), however, we are using this experiment to study protein accumulation at the new SPB. Because cells released from the *CDC20* arrest in a highly synchronized manner, we know we are examining the earliest duplication structures and comparing equivalent intermediates. In addition, its use does not rely on any preconceived ideas about the hierarchical order of SPB assembly that we may have, so it is ideal to ask the basic question: what protein is recruited first, in an unbiased manner?

However, to alleviate concerns over the possibility of a build-up of proteins due to the arrest, we have gone back and quantified images from unbudded cells in an asynchronous population, so it is clear that the temporal order of assembly (shown in Figure 5 and Figure 5—figure supplement 2) is not an artifact of the metaphase arrest. These data, which are in a new Figure 5—figure supplement 2, are consistent with our data from the *CDC20* release, suggesting the Spc42 is the first of the known satellite components to be assembled.

The satellite assembly experiment in Figure 5 and Figure 5—figure supplement 2 is the only place the *CDC20* release was used. We have revised the text to ensure how cells were grown and arrested throughout the paper to make this clear.

*3) The description of the methods is often not sufficient to understand the details. For example, while it is clear how the y-axis is determined, it remains obscure how the authors establish the x-axis. Here, more explanation is required*.

We agree that we could have explained orientation of axes more thoroughly. In Figure 3, the x-axis is defined by Spc42, with the 0 position being the center between the mother and satellite foci. The y-axis 0 position is the center of Spc42, which based on EM, is thought to reside above the nuclear membrane in an intermediate layer of the SPB known as IL2. Additional detail describing how images were aligned, how these axes were defined and how they relate to SPB structure have been added to the text.

In Figure 8, we have used a similar alignment protocol and for clarity and comparison used Spc42 again as the reference points on both axes.

*4) For the position of proteins from the single particle averaging, the distribution of maximum intensity is provided, with the standard error of the mean indicated*.

Throughout the paper, we have provided average intensity distributions as images and in plots rather than maximum intensity distributions because we didn’t want to make assumptions about protein localization variability versus the underlying size in the SPB substructure. We have added text to clarify this in the paper.

*The authors need to include a table indicating additional structural information. This should include the FWHM measurement for each protein. Note in*
Figure 3
*that the distribution of Sfi1 is greater (in width) than Kar1. It is important to have a table with these statistics. In addition, the area over which these proteins occupy could be expressed as the area in which 95% of fluorescence is contained. This would help the reader appreciate the distribution, perhaps with more insight than simply the maximum intensity*.

We have provided full-width half-maximum values and errors for each protein in Table 1 (Figure 3) and Table 2 (Figure 8). As we indicate in the note to each table, these can be converted into 95% confidence values since the distributions are approximately bell-shaped. These results, as well as Figure 3, show that the distribution of Kar1 is greater than Sfi1. We thank the reviewers for this suggestion, and believe inclusion of these numbers will aid those using the new structural information from this manuscript for modeling and design of future experiments.

*5) The manuscript lacks precision at several places and the authors should carefully edit their Results section. For example, in*
Figure 5
*around 70% of pheromone arrested cells show two Nud1 foci (SPB and satellite), whereas in*
Figure 5*, this number drops to less than 10%*. *Why? Is this due to a different duration of pheromone arrest?*

We understand how these differences led to confusion. We believe that time and fluorophores are the key factors leading to this difference. We have acknowledged and discussed the differences between these datasets in the revised text and added an additional time point that illustrates this to Figure 5—figure supplement 2. If we release cells from *CDC20* into α-factor for 3 hours, as we did in Figure 5, 72% of cells show two Nud1 and two Spc42 foci. We thank the reviewers for suggesting we include this data.

*The authors should give this duration for 5A and B either in the main text or in the figure legend. Also, in the main text, the description of the data in*
Figure 5
*does not mention that the cells were arrested with pheromone, which makes the reading confusing*.

We apologize for this oversight. We have revised the text and legends to make it clear how cells were grown and/or arrested. The Materials and methods list the arrest time, temperature, media and amounts of pheromone used in all of our experiments.

Furthermore in 5A and B, while Cnm67 is thought to link Nud1 to the SPB, it is absent from the satellite in more than 60% of the cells where Nud1 is present around 70% of the time. What do the authors make of that?

We have two lines of thinking, which we added to the Discussion.

Based on data from Adams and Kilmartin (JCB, 1999) showing that Nud1-GFP localizes to the SPB in some cells lacking *CNM67* and the fact that *cnm67∆* cells are viable, it is reasonable to consider the idea that Nud1 may associate with the SPB in a Cnm67-independent manner. This likely involves the half-bridge where Spc72 binds, but to our knowledge, no physical evidence places Nud1 here except its ability to associate with Spc72. Nud1 has no known function in SPB duplication, but instead serves as a signaling platform for the mitotic exit network.

However, we also fully acknowledge that detection of Cnm67 at the satellite is challenging and varied considerably from strain to strain. Thus, it is entirely possible that we are underestimating the amount of Cnm67 at the satellite. The same difficulties were also previously reported using anti-Cnm67 antibodies and immuno-EM (Adams and Kilmartin, JCB, 1999).

*There are many examples of similar issues*.

We have carefully scrutinized our text and made changes in the revised version to ensure that the text accurately reflects our data and the published literature.

*In*
Figure 4*, the vast majority of non-arrested cells show a single SPC42 focus associated with a single Sfi1 focus. Does that mean that ∼90% of these cells are in metaphase? In the* mps1-1 *arrested cells, cells show many more configurations*. *Does it mean that the cells arrest at different points in the duplication cycle?*

We apologize for our lack of clarity in the figure legend and text. We have included only large budded cells that have not gone into anaphase in our analysis of wild-type and the mutant at the permissive temperature, since, as the reviewers indicate, these would be the most relevant/comparable to the arrested *mps1-1* cells. We have revised the text and figure legend to ensure this is clear.

*In*
Figures 2 and 3*, the authors suggest the existence of a 17° kink in the bridge. Along which axis is this kink? Could this be simply due to the fact that the bridge is on the membrane while Spc42 is inside the pore?*

Bending occurs in the y-axis, which we have defined as the pole axis that is perpendicular to the nuclear membrane. Based on the distance between Spc42 or Sfi1 foci, and assuming that the G1 haploid nucleus is approximately 1µm in diameter, the natural curvature in the membrane would result in a ∼7° bend in the elongated bridge. Since the 17° bend we observed is greater, we believe there is additional bending. We have added the predicted position of the Sfi1 C-terminus to our plot of bridge coordinates in Figure 3. Based on immunoEM done by the Kilmartin lab, we don’t think satellite-containing Spc42 is inside the membrane in pheromone treated cells. In the revised text, we have discussed this interesting point.